# NPN: Non-Linear Projections of the Null-Space for Imaging Inverse Problems

**Roman Jacome**[†][*]**, Romario Gualdrón-Hurtado**[‡][*]**, Leon Suarez**[‡]**, Henry Arguello**[‡]
[†]Department of Electrical, Electronics, and Telecommunications Engineering
[‡]Department of Systems Engineering and Informatics
Universidad Industrial de Santander, Colombia, 680002
{rajaccar,yesid2238324,leon2238325}@correo.uis.edu.co, henarfu@uis.edu.co

## Abstract

Imaging inverse problems aim to recover high-dimensional signals from undersampled, noisy measurements, a fundamentally ill-posed task with infinite solutions in the null-space of the sensing operator. To resolve this ambiguity, prior information is typically incorporated through handcrafted regularizers or learned models that constrain the solution space. However, these priors typically ignore the task-specific structure of that null-space. In this work, we propose *Non-Linear Projections of the Null-Space* (NPN), a novel class of regularization that, instead of enforcing structural constraints in the image domain, promotes solutions that lie in a low-dimensional projection of the sensing matrix's null-space with a neural network. Our approach has two key advantages: (1) Interpretability: by focusing on the structure of the null-space, we design sensing-matrix-specific priors that capture information orthogonal to the signal components that are fundamentally blind to the sensing process. (2) Flexibility: NPN is adaptable to various inverse problems, compatible with existing reconstruction frameworks, and complementary to conventional image-domain priors. We provide theoretical guarantees on convergence and reconstruction accuracy when used within plug-and-play methods. Empirical results across diverse sensing matrices demonstrate that NPN priors consistently enhance reconstruction fidelity in various imaging inverse problems, such as compressive sensing, deblurring, super-resolution, computed tomography, and magnetic resonance imaging, with plug-and-play methods, unrolling networks, deep image prior, and diffusion models.

## 1 Introduction

Inverse problems involve reconstructing an unknown signal from noisy, corrupted, or undersampled observations, making the recovery process generally non-invertible and ill-posed. This work focuses on linear inverse problems of the form $\mathbf{y} = \mathbf{H}\mathbf{x}^* + \boldsymbol{\omega} \in \mathbb{R}^m$, where $\mathbf{x}^* \in \mathbb{R}^n$ is the target high-dimensional signal, $\mathbf{H} \in \mathbb{R}^{m \times n}$ is the sensing matrix (with $m \ll n$), $\mathbf{y} \in \mathbb{R}^m$ represents the low-dimensional measurements, and $\boldsymbol{\omega} \sim \mathcal{N}(0, \sigma^2 \mathbf{I})$ is additive Gaussian noise. Numerous imaging tasks rely on this principle, including image restoration—such as deblurring, denoising, inpainting, and super-resolution (SR) [21] (structured Toeplitz sensing matrices)—as well as compressed sensing (CS) [61, 4] (dense sensing matrices) and medical imaging applications like magnetic resonance imaging (MRI) [36] (undersampled Fourier matrices) or computed tomography (CT) [59] (Radon matrices). The challenge on the recovery task lies in the ill-posed nature of the inverse problem due to the non-trivial null-space of the sensing matrix $\mathbf{H}$ leading to infinite solutions. Therefore, there is a need to incorporate a signal prior to the reconstruction framework. Under this idea, variational approaches formulate the signal estimator as

$$\hat{\mathbf{x}}_0 = \arg\min_{\tilde{\mathbf{x}}} g(\tilde{\mathbf{x}}) + \lambda h(\tilde{\mathbf{x}}) \tag{1}$$

where $g(\cdot)$ is the data fidelity term and $h(\cdot)$ is a regularization function based on some prior of $\mathbf{x}^*$.

---

[*]Equal contribution.

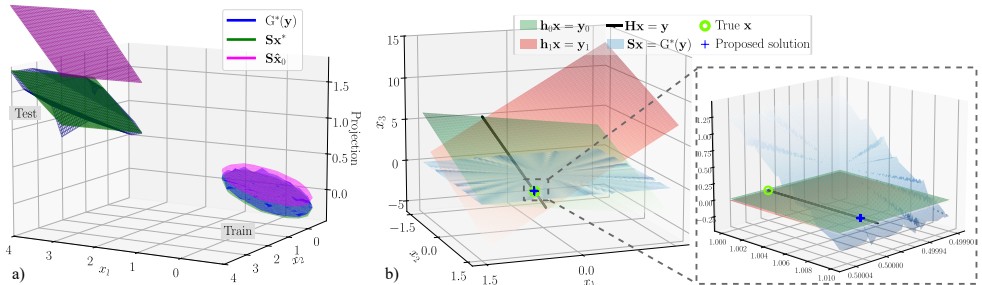

Figure 1: Geometric comparison of subspace–prior learning versus direct reconstruction in a $\mathbb{R}^3$ toy example. **(a)** In the low–dimensional projection space, the learned mapping $G^*(y)$ trained on points inside the unit circle, closely matches the true null–space projection $Sx^*$ for both training (solid) and test (semi-transparent) inputs, whereas the direct–reconstruction estimate $\tilde{x}_0$ projected into $S$ is significantly inaccurate. **(b)** In the original signal domain, the measurements $Hx^* = y$ define two intersecting planes.

One of the most common priors in imaging inverse problems is sparsity, which assumes that images are compressible in a given basis [4]. Other widely used priors include low-rank structures [17] and smoothness priors [19]. Additionally, plug-and-play (PnP) priors [55], which traces back its roots from proximal algorithms [40], where these operators, usually defined by analytical models of the underlying signals [61], are replaced by a general-purpose image denoiser [51]. This approach allows the integration of classical image denoiser [37] such as BM3D [14], NLM [2], RF [16] and current deep learning (DL) denoisers [62]. The idea behind DL-based denoisers is to train a deep neural network (DNN) that maps from the noisy observation to the clean image [3, 53, 52, 24, 23]. Another learning-based approaches are based on the null-space of $H$ by embedding the sensing operator's structure directly into learned networks. In particular, Null-Space Networks exploit the decomposition of a signal into measurement and null-space components, learning a corrective mapping over all null-space modes to enhance interpretability and accuracy [45]. To improve robustness to measurement noise, [6] introduced separate range-space and null-space networks that denoise both components before recombination. Variants of this range–null decomposition have been applied in diffusion-based restoration [11, 57, 58], GAN-prior methods [56], algorithm-unrolling architectures [5], and self-supervised schemes [7, 10], consistently leveraging the full null-space projector to achieve high-fidelity reconstructions.

However, existing learned priors typically promote reconstructions that lie within the subspace spanned by clean training data, without explicitly accounting for the null-space of the sensing matrix $H$. While the data fidelity term $g(\tilde{x})$ enforces consistency with the measurements, it does not sufficiently constrain the null-space components of the solution, especially in the presence of noise, often resulting in suboptimal reconstructions. In this work, we introduce a novel class of regularization, termed *Non-Linear Projections of the Null-Space* (NPN), which directly promotes solutions within a low-dimensional subspace of the null-space of $H$ that is, within the space of vectors orthogonal to the rows of $H$. Our method identifies a compact subspace of the null-space by selecting only the most informative directions and trains a neural network to predict their coefficients directly from measurements. By restricting corrections to this learned subspace, we concentrate regularization on unobserved features most predictive of the true signal. This subspace plays a critical role in addressing the ill-posedness of inverse problems, where conventional methods often struggle due to the lack of constraints in directions invisible to the measurement operator. To enable this, we design a projection matrix $S$ whose rows lie in the null-space of $H$, constructed using either orthogonalization techniques or analytical designs depending on the structure of $H$. A neural network is trained to estimate this null-space projection from the measurements $y$, providing a non-linear prior that is both data-adaptive and model-aware. We further propose a joint optimization framework in which both the projection matrix $S$ and the network are learned simultaneously, allowing the projection matrix to adapt during training in a task-specific manner.

Our approach offers two key advantages for imaging inverse problems: interpretability and flexibility. *Interpretability*: By leveraging a non-linear neural network $G^*$ to learn a projection onto a low-dimensional subspace within the null-space of $H$, we focus on recovering signal components that lie outside the range of the sensing matrix. Learning this projection, rather than directly estimating the full signal $x^*$, reduces complexity through dimensionality reduction while maintaining a clear connection to the geometry of the inverse problem. *Flexibility*: the learned prior can be readily incorporated into a wide variety of reconstruction algorithms and image priors that can be adapted to other imaging inverse problems. To illustrate the interpretability advantage, Figure 1(a) shows

that the learned non-linear estimator $G^*$ closely matches the true null-space projection $\mathbf{Sx}^*$ (green surface), while the projection of the direct reconstruction $\tilde{\mathbf{x}}_0$ onto the same subspace spanned by $\mathbf{S}$ results in significant errors. Moreover, when evaluating out-of-distribution samples within a $2 \times 2$ grid in the range $[2, 4]^2$, our method experiences only a minor increase in estimation error, while the direct reconstruction approach drastically amplifies these errors. In Figure 1(b), we demonstrate that integrating our non-linear subspace prior into the inverse reconstruction process effectively regularizes the solution space. The learned subspace helps uniquely resolve the inverse problem, providing a solution close to the true signal $\mathbf{x}^*$. A zoomed zone near the true solution is shown, where there is a small estimation error with respect to the true value; this is due to the inherent network error in the true subspace. Note that here we did not use any prior $h(\tilde{\mathbf{x}})$, which can reduce the estimation error. See the Appendix A.1 for more details on the setting to obtain this example. We develop a theoretical analysis showing benefits in the convergence rate when the NPN regularizer is used in PnP algorithms. The theory states that the algorithms have a significant speed-up (with respect to the non-regularizer counterpart) in *convergence improvement zone* (CIZ), which takes into account the inherent estimation error of the learned prior. Additionally, we showed that the NPN regularizer is minimized up to a constant depending on the error of the learned prior if the algorithm reaches optimum values. The theory is validated in a comprehensive evaluation of the method in five imaging inverse problems: CS, MRI, deblurring, SR, and CT. Although our theoretical findings are based on PnP methods, we also validated the NPN regularization in unrolling networks, deep image prior, and diffusion models.

## 2 Related work

### 2.1 Variational reconstructions methods

Variational methods solve (1) typically via proximal-gradient schemes that alternate a gradient step on the data fidelity term, $g(\tilde{\mathbf{x}}) = \|\mathbf{H}\tilde{\mathbf{x}} - \mathbf{y}\|_2^2$, with a proximal step for $h$. A classical choice is Tikhonov regularization for smoothness, $h(\mathbf{x}) = \|\mathbf{Lx}\|_2^2$, where $\mathbf{L}$ is a derivative operator or identity to penalize energy and ensure well-posedness [19]. Alternatively, sparsity priors use an $\ell_1$-penalty in a transform domain (e.g. wavelets) to promote compressibility of $\mathbf{x}$ [4]. Algorithms such as ISTA and its accelerated variant FISTA have been widely used to solve $\ell_1$–penalized reconstructions with provably faster convergence rates [1]. More recently, PnP replaces the proximal operator of $h$ with a generic denoiser $\tilde{\mathbf{x}}^k = D_\sigma(\tilde{\mathbf{x}}^{k-1} - \alpha\mathbf{H}^\top(\mathbf{H}\tilde{\mathbf{x}}^{k-1} - \mathbf{y}))$, thus, leveraging sophisticated image priors without an explicit analytic penalty [55, 62]. PnP with both classical denoisers and deep-learning models achieves state-of-the-art results, yet it leaves the null-space of $\mathbf{H}$ uncontrolled: any component in the null-space of $\mathbf{H}$ may be arbitrarily modified by the denoiser.

### 2.2 Null-space learned reconstruction

Harnessing the sensing model into a learning-based reconstruction network enables more accurate reconstructions [45]. Particularly, null-space networks (NSN) [45] harness the range-null-space decomposition (RNSD), which states that a vector $\mathbf{x} \in \mathbb{R}^n$ is decomposed as $\mathbf{x} = P_r(\mathbf{x}) + P_n(\mathbf{x})$ where $P_r(\mathbf{x}) = \mathbf{H}^\dagger\mathbf{Hx}$, with $\mathbf{H}^\dagger$ denoting the Moore–Penrose pseudoinverse, is the projection onto the range space of $\mathbf{H}$ and $P_n(\mathbf{x}) = (\mathbf{I} - \mathbf{H}^\dagger\mathbf{H})\mathbf{x}$ is the null-space projection operator. NSN exploits this structure by learning a neural network $R : \mathbb{R}^n \to \mathbb{R}^n$ such that the reconstruction becomes $\hat{\mathbf{x}} = \mathbf{H}^\dagger\mathbf{y} + (\mathbf{I} - \mathbf{H}^\dagger\mathbf{H})R(\mathbf{H}^\dagger\mathbf{y})$. This approach benefits the interpretability of the reconstruction. However, this method does not take into account the inherent noise of the measurements; thus, [6] introduces deep decomposition networks (DDN), a more robust formulation is presented where two models $R_r$ and $R_n$ are employed to improve recovery performance. Consider two architectures, DDN-independent (DDN-I) $\hat{\mathbf{x}} = \mathbf{H}^\dagger\mathbf{y} + P_r(R_r(\mathbf{H}^\dagger\mathbf{y})) + P_n(R_n(\mathbf{H}^\dagger\mathbf{y}))$ and DDN-Cascade (DDN-C) $\hat{\mathbf{x}} = \mathbf{H}^\dagger\mathbf{y} + P_r(R_r(\mathbf{H}^\dagger\mathbf{y})) + P_n(R_n(\mathbf{H}^\dagger\mathbf{y} + P_r(R_r(\mathbf{H}^\dagger\mathbf{y}))))$. The range-null-space decomposition has also been used to enhance data fidelity in diffusion models for image restoration [11, 57, 58], generative adversarial network priors [56], unfolding networks [5], and self-supervised learning [7, 10]. In these works, the sensing matrix $\mathbf{H}$ structure is incorporated into the reconstruction network through a learned-based RNSD that allows high-fidelity reconstructions. Different from these approaches, our method does not apply the full null-space projection operator. Instead, we first identify a compact subspace of $\mathrm{Null}(\mathbf{H})$ by selecting a projection matrix $\mathbf{S} \in \mathbb{R}^{p \times n}$ whose rows span the most informative null-space directions. We then train a network $G^* : \mathbb{R}^m \to \mathbb{R}^p$ to predict the

coefficients $\mathbf{y}_s = \mathbf{S}\mathbf{x}$ directly from the measurements. On the other hand, the learned NSN [45] is optimized to improve the recovery performance of a determined regularized inverse problem (i.e., Tikhonov-based solutions), which makes it algorithm-specific and does not work as a plug-in for other recovery methods. In our case, since we optimize the network offline, only with the knowledge of $\mathbf{H}$ and $\mathbf{S}$, it can be easily integrated in a wide range of recovery methods.

## 3 Method

In our approach, we design learned priors promoting solutions in a low-dimensional subspace of the null-space of $\mathbf{H}$. First, let's define some useful properties.

**Definition 1** (Null-Space). *The null-space of a matrix $\mathbf{H}$ is defined as*

$$\text{Null}(\mathbf{H}) = \{\mathbf{x} \in \mathbb{R}^n : \mathbf{H}\mathbf{x} = 0\} = \{\mathbf{x} : \mathbf{x} \perp \mathbf{h}_j, \forall j \in \{1, \dots, m\}\}.$$

Thus, we consider a projection matrix $\mathbf{S} \in \mathbb{R}^{p \times n}$, with $p \leq (n-m)$, with rows orthogonal to $\mathbf{H}$ rows, implying that $\mathbf{s}_i \perp \mathbf{h}_j \ \forall i \in \{1, \dots, p\}, \forall j \in \{1, \dots, m\}$. Based on Def. 1, $\mathbf{s}_i \in \text{Null}(\mathbf{H})$ meaning that any projection $\mathbf{y}_s = \mathbf{S}\mathbf{x}$ lies onto a low-dimensional subspace of the null-space of $\mathbf{H}$. Based on this observation, we propose to learn a data-driven prior $G(\cdot)$ restricted to the low-dimensional null-space of $\mathbf{H}$, such that $G(\mathbf{y}) \equiv \mathbf{S}\mathbf{x}^*$. Specifically, we select a projection matrix $\mathbf{S} \in \mathbb{R}^{p \times n}$ whose rows span a subspace of $\text{Null}(\mathbf{H})$. Consequently, we solve

$$\hat{\mathbf{x}} = \arg\min_{\tilde{\mathbf{x}}} \ g(\tilde{\mathbf{x}}) + \lambda h(\tilde{\mathbf{x}}) + \gamma \overbrace{\|G^*(\mathbf{y}) - \mathbf{S}\tilde{\mathbf{x}}\|_2^2}^{\phi(\tilde{\mathbf{x}})}, \tag{2}$$

and $G^* : \mathbb{R}^m \to \mathbb{R}^p$ is a neural network trained to map the measurements $\mathbf{y}$ into the low-dimensional subspace $\mathbf{S}\mathbf{x}^*$, $\gamma$ is a regularization parameter, the regularizer $\phi$ aims to promote solutions on the learned manifold in the null-space of $\mathbf{H}$. Our framework introduces a novel regularization strategy that embeds data-driven models into inverse-problem solvers by constraining solutions to the nonlinear low-dimensional manifold induced by $G^*$, in contrast to existing learned priors that restrict reconstructions to the range of a pre-trained restoration or denoising network [52, 24, 29]. Note that we used the Euclidean norm in the regularizer; however, since the network $G^*$ has some error with respect to $\mathbf{S}\mathbf{x}^*$, one could use a more robust function such as the Huber loss or a weighted norm. Nevertheless, in our experiments, the Euclidean norm works well by adjusting the hyperparameter $\gamma$. One interpretation of the proposed regularization is that it improves the data-fidelity term $g(\tilde{\mathbf{x}})$ as it promotes low-dimensional projections of *blind* signal features to $\mathbf{H}$. Our approach is closely related to NSN-based methods [6, 45, 58], as those methods aim to regularize deep learning-based recovery networks, harnessing the RNSD. In our case, making analogy with these models, we can view the reconstruction in (2) as $\tilde{\mathbf{x}} = \mathbf{H}^\dagger \mathbf{y} + \gamma \mathbf{S}^\dagger G(\mathbf{y}) + \text{prior}_{h,\lambda}$ where the projection onto the null-space $P_n(\cdot)$ is replaced by the range-space $\mathbf{S}$ which promotes solution lying in the $p \leq (n-m)$ most informative null-space modes instead of the entire null-space operator.

### 3.1 Design of the matrix $\mathbf{S}$

To design the matrix $\mathbf{S}$, it is necessary to analyze the structure of $\mathbf{H}$ depending on the inverse problem. Below are insights based on the sensing matrix structure for exploiting the null-space in our prior.

**Compressed Sensing (CS):** In CS, the sensing matrix $\mathbf{H} \in \mathbb{R}^{m \times n}$ is typically dense and randomly generated. Previous approaches [20, 38] use the remaining $(n-m)$ rows of a full-rank Hadamard, Gaussian, or Bernoulli matrix as $\mathbf{S}$. Due to the lack of inherent structure in such matrices, we adopt an orthogonalization strategy based on the classical QR decomposition for designing the matrix $\mathbf{S}$ (see Alg. 1 in Appendix A.2).

**Magnetic Resonance Imaging (MRI):** In MRI, the forward operator $\mathbf{H}$ corresponds to a discrete 2D discrete Fourier transform (DFT) undersampled, where only a subset of frequency components (k-space lines) is acquired during the scan. Specifically, let $\mathcal{F} = \{\mathbf{f}_1^\top, \dots, \mathbf{f}_n^\top\}$ denote the full set of $n$ orthonormal rows of the 2D DFT matrix. The sensing matrix $\mathbf{H}$ then consists of $m < n$ selected rows, i.e., $\{\mathbf{h}_1^\top, \dots, \mathbf{h}_m^\top\} := \mathcal{F}_h \subset \mathcal{F}$. These rows define the measurements taken in the Fourier domain. To construct the null-space projection matrix $\mathbf{S}$, we exploit the fact that the remaining rows in $\mathcal{F}$—those not used in $\mathbf{H}$—span the null-space of the sampling operator. Thus, we define $\{\mathbf{s}_1^\top, \dots, \mathbf{s}_p^\top\} := \mathcal{F}_h^c$, where $\mathcal{F}_h^c = \mathcal{F} \setminus \mathcal{F}_h$ is the complement of the sampled frequencies. Because the DFT matrix is orthonormal, these complementary rows are guaranteed to be orthogonal to the measurement space and form a natural basis for the null-space of $\mathbf{H}$.

**Computed Tomography (CT):** In parallel–beam limited–angle CT, the forward operator samples the Radon transform only at a subset of projection angles. Let $\Theta$ be the full discrete angle set, $\Theta_h \subset \Theta$ the acquired angles, $\mathbf{R}$ is the discrete Radon transform matrix with rows indexed by $\Theta$, and $\mathbf{H} = \mathbf{P}_{\Theta_h}\mathbf{R}$ the forward operator; we define $\mathbf{S}$ directly as the complement of the acquired angles, $\mathbf{S} = \mathbf{P}_{\Theta_h^c}\mathbf{R}$ with $\Theta_h^c = \Theta \setminus \Theta_h$, so $\mathbf{S}$ stacks the rows of $\mathbf{R}$ corresponding to the non-acquired angles.

**Structured Toeplitz matrices (Deblurring and Super-Resolution):** The forward model $\mathbf{H}$ is built upon a Toeplitz matrix based on the convolution kernel denoted as $\mathbf{H}[i, i+j] = \mathbf{h}[j]$ with $i = 1, \ldots, m$ and $j = 1, \ldots, n$. The action of $\mathbf{H}$ corresponds to a linear filtering process, attenuating high-frequency components. From a frequency point of view, the matrix $\mathbf{S}$ should block the low frequencies sampled by $\mathbf{H}$. Thus, we design the matrix $\mathbf{S}$ as $\mathbf{S}[i, i+j] = 1 - \mathbf{h}[j]$ with $i = 1, \ldots, m$ and $j = 1, \ldots, n$.. In super-resolution, the sensing matrix is $\mathbf{H} = \mathbf{DB}$ where $\mathbf{B} \in \mathbb{R}^{n \times n}$ is a convolution matrix build with a low-pass filter $\mathbf{b}$ and $\mathbf{D} \in \mathbb{R}^{m \times n}$ is a decimation matrix denoting $\sqrt{\frac{n}{m}}$ the super-resolution factor (SRF). We construct the matrix $\mathbf{S}$ similarly to the deblurring case, i.e., $\mathbf{S}[i, i+j] = 1 - \mathbf{b}[j]$ with $i = 1, \ldots, m$ and $j = 1, \ldots, n$.

### 3.2 Learning the NPN Prior

Given the design of the projection matrix $\mathbf{S}$, which spans a structured low-dimensional subspace orthogonal to the measurement operator $\mathbf{H}$, we train the network $\mathrm{G}$ to estimate the null-space component $\mathbf{x}^*$ of the signal. To further improve the representation power of the NPN prior, we propose to jointly optimize the neural network $\mathrm{G}$ and the projection matrix $\mathbf{S}$. While the initial design of $\mathbf{S}$ ensures that its rows lie in the null-space of the measurement operator $\mathbf{H}$, optimizing $\mathbf{S}$ during training allows the model to discover a task-adaptive subspace that best complements the measurements. This formulation still preserves the orthogonality between $\mathbf{H}$ and $\mathbf{S}$, while improving the quality of the previously learned. The optimization objective is

$$\mathrm{G}^*, \mathbf{S} = \underset{\mathrm{G}, \tilde{\mathbf{S}}}{\arg\min} \, \mathbb{E}_{\mathbf{x}^*, \mathbf{y}} \|\mathrm{G}(\mathbf{H}\mathbf{x}^*) - \tilde{\mathbf{S}}\mathbf{x}^*\|_2^2 + \lambda_1 \|\mathbf{x}^* - \mathbf{A}^\dagger \mathbf{A}\mathbf{x}^*\|_2^2 + \lambda_2 \|\mathbf{A}^\top \mathbf{A} - \mathbf{I}\|_2^2, \quad (3)$$

where $\mathbf{A} = [\mathbf{H}^\top, \tilde{\mathbf{S}}^\top]^\top$, $\lambda_1, \lambda_2 > 0$ control the trade-off between estimation accuracy and orthogonality enforcement. The first term trains $\mathrm{G}$ to predict the projection $\mathbf{Sx}$ from the measurements $\mathbf{y}$, following the MMSE objective. The second term enforces near-orthogonality between the row spaces of $\mathbf{S}$ and $\mathbf{H}$ by pushing $\mathbf{S}^\top \mathbf{S} + \mathbf{H}^\top \mathbf{H} \approx \mathbf{I}$. The third term promotes full-rank behavior and numerical stability of the combined system matrix $\mathbf{A}$, preventing collapse or redundancy in the learned subspace. Importantly, this formulation enables end-to-end learning of a null-space–aware regularizer, where the matrix $\mathbf{S}$ is initialized using principled designs (e.g., QR orthogonalization or frequency complements in MRI), but is then refined during training to maximize consistency with the true signal statistics and the learned estimator $\mathrm{G}$. This formulation has great benefits in cases of non-structured or dense matrices, such as those in CS; however, for well-defined matrices, such as Fourier-based or Toeplitz matrices, finding orthogonal complements has a straightforward analytical solution, and it is not required to jointly optimize it.

## 4 Theoretical analysis

We analyze how the convergence property of a PnP algorithm is affected under this new regularization. Without loss of generalization, we focus on one of the most common PnP approaches, which is based on proximal gradient methods. The iterations are given by

$$\tilde{\mathbf{x}}^{\ell+1} = \mathrm{T}(\tilde{\mathbf{x}}^\ell) := \mathcal{D}_\sigma \left( \tilde{\mathbf{x}}^\ell - \alpha \left( \nabla g(\tilde{\mathbf{x}}^\ell) + \nabla \phi(\tilde{\mathbf{x}}^\ell) \right) \right), \quad (4)$$

where $\mathcal{D}_\sigma(\cdot)$ is the denoiser operator and $\sigma$ is an hyperparameter modeling the noise variance and $\ell = 1, \ldots, L$ with maximum number of iterations $L$.

Based on this formulation, we analyzed two main aspects: (i) the convergence rate of the algorithm, and (ii) the convergence behavior of the regularization function $\phi(\tilde{\mathbf{x}}^{\ell+1}) = \|\mathrm{G}^*(\mathbf{y}) - \mathbf{S}\tilde{\mathbf{x}}^{\ell+1}\|_2^2$. Our theoretical developments leverage the restricted isometry property defined over a specific Riemannian manifold $\mathcal{M}_\mathcal{D}$, induced by the image-space of the denoiser $\mathcal{D}_\sigma$. For the denoiser to exhibit isometric-like properties, certain criteria must be satisfied, including boundedness, Lipschitz continuity, and a low-rank Jacobian. These properties can be guaranteed through spectral normalization during denoiser training [43]. Such assumptions are commonly employed in the convergence analysis of

iterative projection methods [46] and have been suitably adapted for PnP convergence analyses [31]. Additionally, we introduce assumptions regarding the estimation error of the model $G^*$, assuming a Gaussian error distribution. Furthermore, we define a CIZ in the algorithm iterations based on the estimation error norm. For the guarantees, we consider the noiseless case, i.e., $\boldsymbol{\omega} = \mathbf{0}$.

**Definition 2** (Restricted Isometry Property [46]). *Let $\mathcal{M}_D \subset \mathbb{R}^n$ be a Riemannian manifold given by the denoiser's image space $\text{Im}(\mathcal{D}_\sigma)$, thus $\mathbf{S} \in \mathbb{R}^{p \times n}$ satisfies the restricted isometry property with respect to $\mathcal{M}_D$ with a restricted isometry constant (RIC) $\Delta^S_{\mathcal{M}_D} \in [0, 1)$ if for all $\mathbf{x}, \mathbf{z} \in \mathcal{M}_D$ with*

$$(1 - \Delta^S_{\mathcal{M}_D})\|\mathbf{x} - \mathbf{z}\|_2^2 \leq \|\mathbf{S}(\mathbf{x} - \mathbf{z})\|_2^2 \leq (1 + \Delta^S_{\mathcal{M}_D})\|\mathbf{x} - \mathbf{z}\|_2^2. \tag{5}$$

**Assumption 1** (Prior mismatch). *The trained model $G^*$ using (3), we assume that*

$$G^*(\mathbf{H}\mathbf{x}^*) = \overbrace{\mathbf{S}\mathbf{x}^*}^{\text{Ground truth value}} + \underbrace{N(\mathbf{H}\mathbf{x}^*)}_{\text{Non-linear error term}}. \tag{6}$$

*Thus, considering that the nonlinear operator $N$ is $K$-Lipschitz continuous, and $\mathbf{H}$ satisfies Definition 2 with a constant $\Delta^H_{\mathcal{M}_D}$, thus we have $\|N(\mathbf{H}\mathbf{x}^*)\|_2^2 \leq K(1 + \Delta^H_{\mathcal{M}_D})\|\mathbf{x}^*\|_2^2$.*

**Definition 3** (Convergence Improvement Zone (CIZ) by $\phi(\mathbf{x}^\ell)$). *We define a zone where the proposed NPN prior improves the convergence of the PnP algorithm. For that, $\mathbf{S}$ satisfies the RIP with RIC $\Delta^S_{\mathcal{M}_D} \in [0, 1)$, the CIZ by $\phi$ are the iterations $\mathcal{L}_\phi = \{1, \ldots, L_\phi\}$ where $L_\phi \leq L$ such that the network estimation error $N(\mathbf{H}\mathbf{x})$ for all $\ell \in \mathcal{L}_\phi$ satisfies that*

$$\|N(\mathbf{H}\mathbf{x})\|_2^2 \leq \|\mathbf{S}\tilde{\mathbf{x}}^\ell - \mathbf{S}\mathbf{x}^*\|_2^2 = \|\mathbf{S}\left(\tilde{\mathbf{x}}^\ell - \mathbf{x}^*\right)\|_2^2 \leq (1 + \Delta^S_{\mathcal{M}_D})\|\tilde{\mathbf{x}}^\ell - \mathbf{x}^*\|_2^2$$

**Assumption 2** (Bounded denoiser). *We consider that for $\mathbf{x}, \mathbf{z} \in \mathbb{R}^n$, $\mathcal{D}_\sigma$ is a bounded denoiser, with a constant $\delta > 0$, if*

$$\|\mathcal{D}_\sigma(\mathbf{x}) - \mathcal{D}_\sigma(\mathbf{z})\|_2^2 \leq (1 + \delta)\|\mathbf{x} - \mathbf{z}\|_2^2$$

We are now equipped to develop the first theoretical benefit of the proposed method.

**Theorem 1** (PnP-NPN Convergence). *Consider the fidelity term $g(\tilde{\mathbf{x}}) = \|\mathbf{y} - \mathbf{H}\tilde{\mathbf{x}}\|_2^2$, and assume the denoiser $\mathcal{D}_\sigma$ satisfies Assumption 2. Let the matrix $\mathbf{S}$ be constructed according to (3) and satisfy the RIP condition (Definition 2) with constant $\Delta^S_{\mathcal{M}_D} \in [0, 1)$. Then, for a finite number of iterations $\ell = 1, \ldots, L_\phi$ within the CIZ, the residual $\|\mathbf{x}^{\ell+1} - \mathbf{x}^*\|_2^2$ decays linearly with rate*

$$\rho \triangleq (1 + \delta)\left(\|\mathbf{I} - \alpha(\mathbf{H}^\top\mathbf{H} + \mathbf{S}^\top\mathbf{S})\|_2^2 + (1 + \Delta^S_{\mathcal{M}_D})\|\mathbf{S}\|_2^2\right) < 1 \tag{7}$$

The proof of the theorem can be found in Appendix A.3. A key insight from Theorem 1 is the role of the CIZ $\mathcal{L}_\phi$, which characterizes the subset of iterations where the proposed regularizer outperforms provides improved convergence. Specifically, due to the inherent mismatch between the ground-truth projection $\mathbf{S}\mathbf{x}^*$ and its learned estimate $G^*(\mathbf{H}\mathbf{x}^*)$, the NPN prior is only effective while the projected estimate $\mathbf{S}\tilde{\mathbf{x}}^\ell$ remains closer to $\mathbf{S}\mathbf{x}^*$ than the residual error $N(\mathbf{H}\mathbf{x}^*)$. Outside this zone, the regularizer may no longer provide beneficial guidance to the reconstruction. Nevertheless, thanks to the design of the matrix $\mathbf{S}$ either through analytical construction or data-driven optimization, it is guaranteed to be orthogonal to the rows of the sensing matrix $\mathbf{H}$. This orthogonality ensures that the operator norm $\|\mathbf{I} - \alpha(\mathbf{H}^\top\mathbf{H} + \mathbf{S}^\top\mathbf{S})\|_2^2$ remains small for an appropriate step size $\alpha$. Furthermore, when the spectral norm of $\mathbf{S}$ and its restricted isometry constant $\Delta^S_{\mathcal{M}_D}$ are sufficiently low, the acceleration factor $\rho$ falls below one. This guarantees that the PnP-NPN algorithm will converge to a fixed point within the zone $\mathcal{L}_\phi$, thereby validating the theoretical and practical benefits of incorporating the NPN prior into the reconstruction framework.

The second analysis of the proposed approach is the convergence of the regularization.

**Theorem 2** (Convergence of NPN Regularization). *Consider the iterations of the PnP-NPN algorithm defined in (4) for $\ell = 1, \ldots, L$. Assume that the estimation error of the trained network $G^*$ satisfies Assumption 1, and that both $\mathbf{S}$ and $\mathbf{H}$ satisfy the Restricted Isometry Property (RIP) over the manifold $\mathcal{M}_D$ with constants $\Delta^S_{\mathcal{M}_D}, \Delta^H_{\mathcal{M}_D} \in [0, 1)$. Further assume that the residual term $N(\mathbf{H}\mathbf{x}^*)$ is $K$-Lipschitz continuous, thus $\|N(\mathbf{H}\mathbf{x}^*)\|_2^2 \leq K(1 + \Delta^H_{M_D})\|\mathbf{x}^*\|_2^2$. Then, after $\ell$ iterations, the NPN regularization term satisfies the following upper bound:*

$$\|G(\mathbf{y}) - \mathbf{S}\tilde{\mathbf{x}}^{\ell+1}\|_2^2 \leq C_1\|\mathbf{x}^* - \mathbf{x}^\ell\|_2^2 + K\|\mathbf{x}^*\|_2^2(1 + \Delta^H_{\mathcal{M}_D}) + C_2\|\mathbf{x}^\ell - \mathbf{x}^{\ell+1}\|_2^2, \tag{8}$$

*where $C_1 = \left(\frac{1}{2\alpha} + K(1 + \Delta^H_{\mathcal{M}_D})\|\mathbf{x}^*\|_2^2\right)(1 + \Delta^S_{\mathcal{M}_D})$, $C_2 = \left(1 + \Delta^S_{\mathcal{M}_D}\right)^2 \left(1 + \frac{1}{2\alpha}\right)$.*

The proof of Theorem 2 is provided in the technical Appendix A.4. This result shows that the regularization value $\phi(\mathbf{x}^{\ell+1})$ decreases as the reconstruction error $\|\mathbf{x}^* - \mathbf{x}^\ell\|_2^2$ diminishes over the course of the iterations. From Theorem 1, we know that $\|\mathbf{x}^* - \mathbf{x}^\ell\|_2^2$ is reduced, thereby ensuring a monotonic decrease in the regularization term. Moreover, since Theorem 1 guarantees convergence of the sequence $\{\mathbf{x}^\ell\}$ to a fixed point, the difference $\|\mathbf{x}^\ell - \mathbf{x}^{\ell+1}\|_2^2$ asymptotically vanishes as $\ell \to \infty$. Therefore, in the limit, the regularization value is bounded above by a residual term determined by the Lipschitz constant $K$, which can be minimized by including spectral normalization in G, the norm of the ground truth signal $\|\mathbf{x}^*\|_2^2$, and the RIP constant of the sensing matrix: $\lim_{\ell \to \infty} \phi(\mathbf{x}^{\ell+1}) \leq K\|\mathbf{x}^*\|_2^2(1 + \Delta_{\mathcal{M}_D}^H)$. This bound quantifies the asymptotic regularization performance of the NPN prior in terms of the approximation quality of the learned model and the sensing operator geometry.

Our theoretical analysis has focused on integrating the proposed NPN regularizer into projected-gradient-descent-based PnP methods. However, the same regularization strategy can be applied to a broad class of learning-based solvers. For example, in algorithm-unrolling architectures [39], deep equilibrium models [18], and learning-to-optimize frameworks [8], one can insert the NPN proximal step alongside the usual network updates, with end-to-end training of all learnable parameters [44, 9]. Although we demonstrate the empirical benefits of NPN within an unrolled network in Section 5, a full theoretical treatment of these extensions is left for future work. We also incorporate the NPN regularization in Deep Image Prior (DIP) [54] framework (see Appendix A.7 for more details) and in two diffusion-based solvers [15], diffusion posterior sampling (DPS) [12], and DiffPIR [65] (see Appendix A.8 for more details).

## 5 Experiments

The proposed NPN regularization was evaluated in five imaging inverse problems: compressed sensing, super-resolution, computed tomography, single coil MRI, and deblurring. The method was implemented using the PyTorch framework. For the recovery algorithm, we adopt the FISTA solver [1], with a deep denoiser [25], regularization by denoising (RED) [41], and sparsity prior [1], see Appendix A.5. All simulations were performed on an NVIDIA RTX 4090 GPU, with code in [1].

**Compressed Sensing:** The single-pixel camera (SPC) is used along with the CIFAR-10 dataset [28], with $50,000$ images for training and $10,000$ for testing. All images were resized to $32 \times 32$. The Adam [26] optimizer was used with a learning rate of $5 \times 10^{-4}$. $\mathbf{H}$ is a random binary sensing matrix with $m/n = 0.1$. $\mathbf{S}$ is initialized by QR decomposition, with Algorithm 1 in Appendix A.2. Then, G (for which we used a ConvNeXt [33]) and $\mathbf{S}$ are optimized following Eq. (3).

**MRI:** We employed the fastMRI knee single-coil MRI dataset [27], which consists of 900 training images and 73 test images of knee MRIs of $320 \times 320$. The training set was split into 810 images for training and 90 for validation, and all images were resized to $256 \times 256$. For G we used a U-Net based architecture [42] and trained it for 60 epochs with a learning rate of $1 \times 10^{-4}$, using the AdamW optimizer [35] with a weight decay of $1 \times 10^{-2}$ and a batch size of 4. Performance was evaluated at acceleration factors ($AF = \frac{n}{m}$) of 4, 8, and 16, using a Radial undersampling mask [32].

**Deblurring and Super-Resolution:** For the deblurring inverse problem, we used a 2-D Gaussian kernel with a variance $\sigma$. For these experiments, we used the CelebA [34] dataset resized to $128 \times 128$, using 8000 images for training and 2000 for testing. Here, we employed a U-Net architecture for the network G. In this case, we used the Adam optimizer with a learning rate of $1 \times 10^{-3}$ and a batch size of 32. The results for the super-resolution case are shown in Appendix A.10.

**Computed Tomography:** We evaluated the proposed regularization on the limited-angle CT inverse problem, using 60 out of 180 total views (spaced every 1°) under a parallel-beam geometry. We train the DM for 1000 epochs with batch size 4 using the AdamW optimizer and learning rate $3 \times 10^{-4}$. We used a cosine variance schedule ranging from $\beta_1 = 1 \times 10^{-4}$ to $\beta_1 = 0.02$ with $T = 1000$ time steps. The LoDoPaB-CT dataset [30] was resized to $256 \times 256$ and used for training; in testing, we used 10 test set slices. G is a U-Net which was trained for 100 epochs with a learning rate $3 \times 10^{-4}$ and a batch size of 4 using AdamW.

---

[1] github.com/yromariogh/NPN

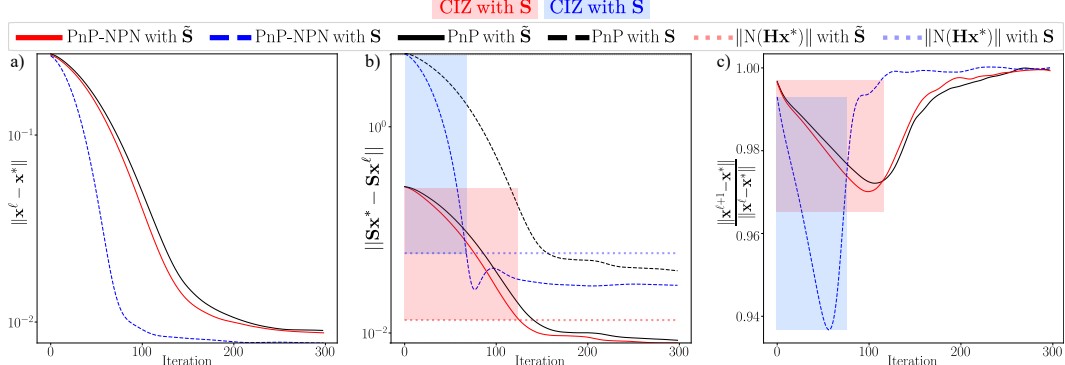

Figure 2: PnP-FISTA convergence analysis in CS. **(a)** Reconstruction error. **(b)** Null-space prediction error for (red) Initialization $\tilde{\mathbf{S}} = \mathrm{QR}(\mathbf{H})$ from Algorithm 1, and (blue) Designed $\mathbf{S}$ with Eq. (3) and $m/n = p/n = 0.1$. In this case, the CIZ from Definition 3 is highlighted in light red and light blue. **(c)** Acceleration ratio of signal convergence; here, the CIZ is defined as the empirical convergence ratio of the proposed solution that is lower than the baseline (black).

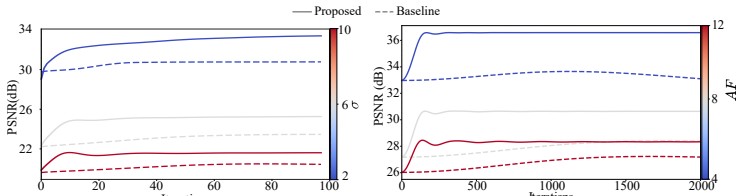

Figure 3: Convergence curves for $\sigma \in \{2, 5, 10\}$ in deblurring, AF $\in \{4, 8, 12\}$ in MRI.

## 5.1 Convergence Analysis

To confirm numerically the theoretical results obtained in Sec. 4, we show Fig. 2 with 3 general cases: Baseline (black), where only $\mathbf{H}$ is used; PnP-NPN with designed $\mathbf{S}$ (blue), where G and $\mathbf{S}$ are jointly trained following Eq. (3), with $\lambda_1 = 0.001$, $\lambda_2 = 0.01$; PnP-NPN with QR (red) where the projection matrix $\tilde{\mathbf{S}} = \mathrm{QR}(\mathbf{H})$ is fixed (for details about $\mathrm{QR}(\cdot)$ see Algorithm 1 in Appendix A.2), only G is trained, and $\lambda_1 = \lambda_2 = 0$. In Fig. 2(a), we plot the error $\mathcal{E}_\ell = \|\mathbf{x}^\ell - \mathbf{x}^*\|_2^2$, showing that both NPN-QR (red) and NPN-designed (blue) decay much faster than the baseline (black) over the first $\sim 75$ iterations. Fig. 2(c) tracks the projection onto null-space $\|\mathbf{S}\mathbf{x}^* - \mathbf{S}\mathbf{x}^\ell\|_2^2$, which steadily decreases where $\|\mathrm{N}(\mathbf{H}\mathbf{x}^*)\|_2^2 \leq \|\mathbf{S}(\mathbf{x}^\ell - \mathbf{x}^*)\|_2^2$, whereas the baseline's reprojection error (dashed black) remains high where. The CIZ is defined as in Def. 3. Fig. 2(c) shows the per-step acceleration ratio $\|\mathbf{x}^{\ell+1} - \mathbf{x}^*\|_2^2 / \|\mathbf{x}^\ell - \mathbf{x}^*\|_2^2$, where both NPN curves dip well below the baseline. In this scenario, the CIZ is defined as the iterations when the convergence ratio of the proposed method is lower than that of the baseline, with the designed $\mathbf{S}$ achieving the smallest $R_\ell$ around $\ell \approx 50$–$75$, confirming stronger per-iteration error reduction. The results show that empirical improvements in the algorithm convergence are predictable with the CIZ validating Theorem 1. Figure 3 shows convergence plots in PSNR for MRI with $AF = \{4, 8, 12\}$, and for deblurring and $\sigma = \{2, 5, 10\}$, where NPN regularization consistently yields higher reconstruction quality and faster convergence compared to the baseline. Additional results with other state-of-the-art denoisers in PnP-ADMM are shown in Appendix A.11 for the image deblurring task. For the CS scenario, the selection $p$ and the joint optimization in Eq. (3) is fundamental, see Appendices A.11 and A.12 for detailed analysis.

## 5.2 Visual results

Figure 4 presents reconstruction results for MRI with an $AF = 4$, CT with an acquisition of 30 views of a total of 180, and for deblurring with $\sigma = 2$ using PnP with the DnCNN prior [63]. The estimate $\hat{\mathbf{x}}_0$ is obtained via equation (1), while the estimate $\hat{\mathbf{x}}$ is obtained with NPN regularization through equation (2). Results show that the learned prior effectively approximates the true nullspace $\mathrm{G}(\mathbf{y}) \approx \mathbf{S}\mathbf{x}^*$ with an estimation PSNR of 28.11 dB in deblurring, 39.13 dB for MRI. Moreover, the reconstruction $\hat{\mathbf{x}}$ obtained with NPN regularization preserves high-frequency details and provides overall improved performance than $\hat{\mathbf{x}}_0$.

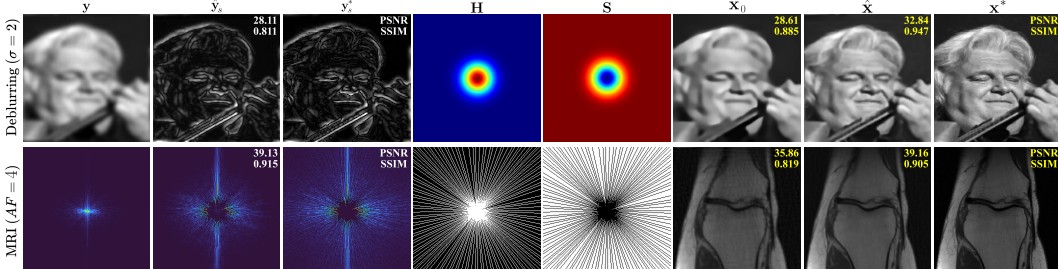

Figure 4: Deblurring and MRI reconstruction results for PnP and PnP-NPN using a DnCNN prior, with 5 dB SNR measurement noise. The measurements are denoted by $\mathbf{y}$, the nonlinear approximation of the null-space of $\mathbf{H}$ by $\hat{\mathbf{y}}_s$, and the true null-space by $\mathbf{y}_s^*$. The estimation $\hat{\mathbf{x}}_0$ is obtained with (1), and $\hat{\mathbf{x}}$ is obtained with the proposed regularization with (2). The ground truth signal is $\mathbf{x}^*$.

| Inverse Problem | Baseline | | | NPN | | | NSN | | |
| --- | --- | --- | --- | --- | --- | --- | --- | --- | --- |
| | Sparsity | PnP | RED | Sparsity | PnP | RED | DNSN [45] | DDN-C [6] | DDN-I [6] |
| CS ($\gamma = 0.1$) | 15.93 | 20.04 | 17.45 | 16.15 | **21.12** | 17.53 | 20.10 | 20.03 | 20.7 |
| MRI (AF = 4) | 36.86 | 35.99 | 36.00 | **38.16** | 38.08 | 38.07 | 35.2 | 33.7 | 33.2 |
| Deblurring ($\sigma = 2$) | 29.27 | 30.78 | 32.84 | 31.77 | 31.42 | **33.67** | 33.07 | 33.03 | 32.70 |

Table 1: State-of-the-art comparison for CS, MRI, and Deblurring. For each task, the best results are highlighted in **bold teal**, while the second-best results are shown in underline orange.

## 5.3 State-of-the-art comparison

We compared our approach with other state-of-the-art recovery methods that exploit the null-space structure. Mainly, we compare with the DNSN [45] and the DDN-I and DDN-C [6] (For more details on the implementation of these methods, refer to Appendix A.9). The results for CS, MRI, and Deblurring are in Table 1, showing consistent improvement or competitive performance of the proposed method. In Appendix A.14, we show a comparison where the same neural network as G, instead of estimating $\mathbf{Sx}$ it estimates directly $\mathbf{x}$, and incorporates it as a regularizer in a PnP-ADMM algorithm, where we show that the proposed method achieves superior performance.

| Method | $p/n$ | CIFAR-10 | | STL10 | |
| --- | --- | --- | --- | --- | --- |
| | | PnP | Unrolling | PnP | Unrolling |
| Baseline | 0.0 | 20.04 | 24.32 | 20.09 | 18.35 |
| NPN | 0.1 | 21.12 | 28.53 | 19.91 | 19.64 |
| | 0.3 | 21.07 | 28.75 | **21.14** | 20.23 |
| | 0.5 | 20.78 | 27.64 | 20.77 | 18.76 |
| | 0.7 | 20.09 | 26.73 | 20.31 | 18.45 |
| | 0.9 | 20.41 | **29.90** | 21.02 | 19.48 |

Table 2: Dataset generalization results for SPC in PnP and Unrolling. Each $\mathbf{S}$, $G^*$, and Unrolling were optimized with CIFAR-10, and tested with the CIFAR-10 and STL10. For each dataset, the best results are highlighted in **bold teal**, while the second-best results are shown in underline orange.

## 5.4 Performance in data-driven models and dataset generalization

In Table 2 we report PSNR (dB) for both PnP and unrolling solvers on CIFAR-10 [28] (in-distribution) and STL10 [13] (out-of-distribution) across projection ratios $p/n$. PnP achieves a 1 dB boost at $p/n = 0.1$ and sustains 1 dB gains on STL10, peaking when $p/n = 0.3$. Unrolling delivers up to 5.6 dB in-distribution improvement but only 1.9 dB cross-dataset gain at $p/n = 0.3$ before declining. Overall, PnP ensures stable generalization, while unrolling maximizes peak PSNR; $p/n \approx 0.3$ provides the best balance between accuracy and robustness. In general, NPN improves both PnP and unrolling reconstruction performance regarding dataset changes.

## 5.5 Deep Image Prior

We consider the deblurring task with a kernel bandwidth of $\sigma = 4.0$. We train G with the Places365 dataset [64], where we used 28.000 images for training and 7000 images for testing. All images were resized to $128 \times 128$. The network K was trained following (15) using the Adam optimizer with a learning rate of $1e^{-3}$ for 1000 iterations. The network K is a U-Net of the same size as G. In Fig. 5 is shown the reconstruction performance of DIP and NPN-DIP for different values of $\gamma$. The results show significant improvements of up to 5 dB and convergence improvements.

## 5.6 Diffusion model solvers

We integrated the proposed regularization term into two widely adopted DM frameworks, DPS [12] and DiffPIR [65]. Table 3 shows the obtained results for different values of $\gamma$. For DPS, the NPN regularization consistently improves reconstruction performance by up to 1.85 dB. For DiffPIR, it yields improvements of up to 0.61 dB. Figure 6 shows reconstruction results using NPN within DPS and DiffPIR; yellow arrows highlight improved image details with NPN. Details on the implementation of NPN into DPS and DiffPIR are provided in the Appendix A.8.

| $\gamma$ | NPN–DPS | NPN–DiffPIR |
|---|---|---|
| 0.0 (Base) | 28.22 | 31.30 |
| $10^{-5}$ | 28.55 | 31.88 |
| $10^{-4}$ | 28.30 | **31.91** |
| 0.1 | 30.06 | 28.98 |
| 0.2 | **30.07** | 28.57 |

Table 3: Ablation over $\gamma$ for two methods. Best results are bold teal ; second-best are underline orange .

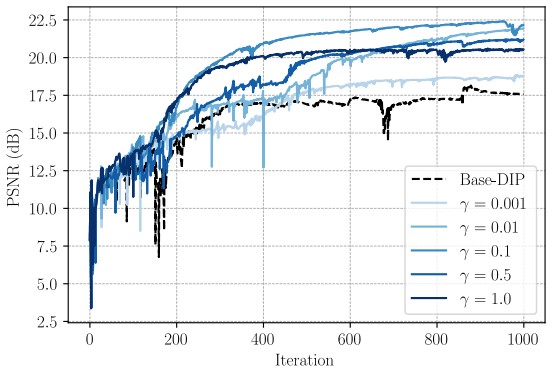

Figure 5: Performance of DIP and NPN-DIP for different values of $\gamma$ in Deblurring.

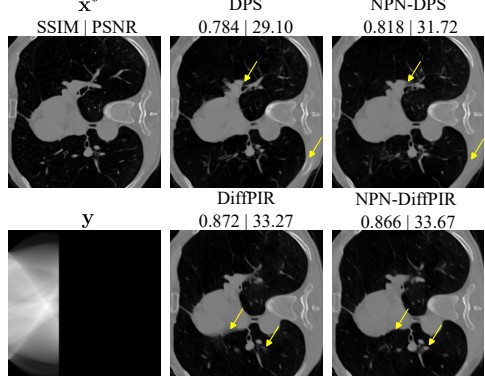

Figure 6: Reconstruction results using NPN within DPS and DiffPIR.

## 6 Limitations

While the proposed method introduces some additional complexity, such as training a dedicated neural network for each sensing configuration $(\mathbf{H}, \mathbf{S})$, the networks are lightweight and tailored to specific inverse problems, keeping computational demands modest (see Appendices A.5 and A.14 for details). The current integration into learning-based reconstruction frameworks (e.g., unrolling networks) involves a two-stage training process, first for the NPN regularizer, then for the reconstruction model, but this modular design enables flexible adaptation. Future work could explore joint end-to-end training of the NPN regularizer and reconstruction network to further enhance performance and efficiency. Unlike NSN-based methods [45, 6], which directly reconstruct, our approach learns a subspace projection offline, adding an extra training step but offering improved compatibility with different solvers. Although we devise some design criteria for selection $\mathbf{S}$, there are scenarios in which the method can fail upon this design, we provide a detailed discussion in A.13 on this aspect from the point of view of finding non-linear relations between $\mathbf{Hx}$ and $\mathbf{Sx}$.

## 7 Conclusion and future outlooks

We introduce *Non-Linear Projections of the Null-Space* for regularizing imaging inverse problems. Intuitively, the regularization promotes selective coefficients of the signal in the null-space of the sensing matrix. This formulation allows flexibility in what features of the null-space we can exploit. Our proposed method is equipped with strong theoretical guarantees for plug-and-play restoration algorithms, showing that the proposed regularization has a zone of convergence improvement controlled by the network error. Additionally, we show that our regularizer converges to a constant depending also on the network estimation error when the algorithm reaches the optimum. We validate our theoretical findings in five distinctive imaging inverse problems: compressed sensing, magnetic resonance, super-resolution, computed tomography, and deblurring. Results validate the theoretical developments, and we have improved performance with state-of-the-art methods. This approach opens new frontiers to regularize the imaging inverse problems in different solvers, such as deep equilibrium models and consensus equilibrium formulations.

## Acknowledgements

This work was supported in part by the Agencia Nacional de Hidrocarburos (ANH) and the Ministerio de Ciencia, Tecnología e Innovación (MINCIENCIAS), under contract 045-2025, and in part by the Army Research Office/Laboratory under grant number W911NF-25-1-0165, VIE from UIS project 8087. The views and conclusions contained in this document are those of the authors and should not be interpreted as representing the official policies, either expressed or implied, of the Army Research Laboratory or the U.S. Government.

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

## A Technical Appendices

### A.1 Settings for experiment in Figure 1

To illustrate this concept, we developed a toy example with $\mathbf{x}^* \in \mathbb{R}^3$, $\mathbf{H} \in \mathbb{R}^{2 \times 3}$, and constructed an orthonormal vector $\mathbf{S} \in \mathbb{R}^{1 \times 3}$ relative to the rows of $\mathbf{H}$, following Algorithm 1. A training dataset $p(\mathbf{x})$ was generated, consisting of random points within a circle centered at the origin. Using this dataset, we trained a two-layer neural network defined as $G(\mathbf{Hx}) = \mathbf{V}\phi(\mathbf{WHx})$, with parameters $\mathbf{W} \in \mathbb{R}^{k \times 2}$ and $\mathbf{V} \in \mathbb{R}^{1 \times k}$, with $k = 50$ hidden neurons, optimized via the problem stated in (3). For comparison, we also trained a network of similar size to directly reconstruct $\tilde{\mathbf{x}}_0$ from measurements $\mathbf{Hx}^*$.

### A.2 Algorithms for designing S

We developed an algorithm for obtaining $\mathbf{S}$ from $\mathbf{H}$ satisfying that $\mathbf{SH}^\top = \mathbf{0}$. The algorithm is based on the QR decomposition, first, computing a full QR decomposition of $\mathbf{H}^\top \in \mathbb{R}^{n \times m}$, yielding an orthonormal basis $\mathbf{Q}_{\text{full}} \in \mathbb{R}^{n \times n}$ for $\mathbb{R}^n$. The columns from $m+1$ to $n$ of $\mathbf{Q}_{\text{full}}$ form a basis for $\text{Null}(\mathbf{H})$, which we denote by $\mathbf{N} \in \mathbb{R}^{n \times (n-m)}$. To construct a subspace of the null space, the algorithm samples a random Gaussian matrix $\mathbf{P} \in \mathbb{R}^{(n-m) \times p}$, which is orthonormalized via QR decomposition to produce $\mathbf{U} \in \mathbb{R}^{(n-m) \times p}$. This ensures that the resulting subspace is both diverse and well-conditioned. Finally, the matrix $\mathbf{S}$ is obtained as $\mathbf{S} = \mathbf{U}^\top \mathbf{N}^\top \in \mathbb{R}^{p \times n}$, which consists of $p$ orthonormal vectors that span a random $p$-dimensional subspace within $\text{Null}(\mathbf{H})$.

---

**Algorithm 1** GENERATE ORTHONORMAL ROWS TO **H** VIA QR DECOMPOSITION

---

**Require:** Matrix $\mathbf{H} \in \mathbb{R}^{m \times n}$, desired number of rows $p$
**Ensure:** Matrix $\mathbf{S} \in \mathbb{R}^{p \times n}$ whose rows are orthonormal and lie in $\text{Null}(\mathbf{H})$
1: $\mathbf{Q}_{\text{full}} \leftarrow \text{QR}(\mathbf{H}^\top)$
2: $\mathbf{N} \leftarrow \mathbf{Q}_{\text{full}}[:, m+1:n]$          ▷ Nullspace basis, size $n \times (n-m)$
3: Sample $\mathbf{P} \sim \mathcal{N}(0, I) \in \mathbb{R}^{(n-m) \times p}$
4: $\mathbf{U} \leftarrow \text{QR}(\mathbf{P})$          ▷ $\mathbf{U} \in \mathbb{R}^{(n-m) \times p}$ with orthonormal columns
5: $\mathbf{S} \leftarrow \mathbf{U}^\top \mathbf{N}^\top$          ▷ Resulting $p \times n$ matrix of orthonormal rows
6: **return S**

---

### A.3 Proof of Theorem 1

Here we provide the technical proof for Theorem 1.

**Theorem 1** (PnP-NPN Convergence). *Consider the fidelity $g(\tilde{\mathbf{x}}) = \|\mathbf{H}\tilde{\mathbf{x}} - \mathbf{y}\|_2^2$, the denoiser $\mathcal{D}_\sigma$ satisfies assumption 2, the matrix $\mathbf{S}$ satisfies the RIP condition 2 with constant $\Delta_{\mathcal{M}_D}^S \in [0,1)$. The optimized network $\mathcal{G}^*$ satisfies assumption 3. We run iterations as in (4) in the convergence improvement zone for $\ell = 1, \ldots, L_\phi$. The algorithm converges to a fixed point if*

$$\rho \triangleq (1+\delta)\left(\left\|\mathbf{I} - \alpha(\mathbf{H}^\top\mathbf{H} + \mathbf{S}^\top\mathbf{S})\right\| + (1 + \Delta_{\mathcal{M}_D}^S)\|\mathbf{S}\|\right) < 1 \tag{9}$$

*Proof:*

$\|\mathbf{x}^{\ell+1} - \mathbf{x}^*\| = \|\text{T}(\mathbf{x}^\ell) - \text{T}(\mathbf{x}^*)\|$

$\overset{\text{Assumption 2}}{\leq} (1+\delta)\left\|\mathbf{x}^\ell - \alpha\left(\mathbf{H}^\top(\mathbf{Hx}^\ell - \mathbf{y}) + \mathbf{S}^\top(\mathbf{Sx}^\ell - (\mathbf{Sx}^* + \mathcal{N}(\mathbf{Hx}^*)))\right) - \mathbf{x}^*\right\|_2$

$= (1+\delta)\left\|\mathbf{x}^\ell - \mathbf{x}^* - \alpha(\mathbf{H}^\top\mathbf{H} + \mathbf{S}^\top\mathbf{S})(\mathbf{x}^\ell - \mathbf{x}^*) + \mathbf{S}^\top\mathcal{N}(\mathbf{Hx}^*)\right\|_2$

$= (1+\delta)\left\|(\mathbf{I} - \alpha(\mathbf{H}^\top\mathbf{H} + \mathbf{S}^\top\mathbf{S}))(\mathbf{x}^\ell - \mathbf{x}^*) + \mathbf{S}^\top\mathcal{N}(\mathbf{Hx}^*)\right\|$

$\overset{\text{triang. ineq.}}{\leq} (1+\delta)\left\|\mathbf{I} - \alpha(\mathbf{H}^\top\mathbf{H} + \mathbf{S}^\top\mathbf{S})\right\|\|\mathbf{x}^\ell - \mathbf{x}^*\| + (1+\delta)\|\mathbf{S}^\top\mathcal{N}(\mathbf{Hx}^*)\|$

$\overset{\text{Definition 3}}{\leq} (1+\delta)\left\|\mathbf{I} - \alpha(\mathbf{H}^\top\mathbf{H} + \mathbf{S}^\top\mathbf{S})\right\|\|\mathbf{x}^\ell - \mathbf{x}^*\| + (1+\delta)\|\mathbf{S}\|\|\mathbf{S}\left(\tilde{\mathbf{x}}^\ell - \mathbf{x}_*\right)\|$

$\overset{\text{Definition 2}}{\leq} (1+\delta)\left(\left\|\mathbf{I} - \alpha(\mathbf{H}^\top\mathbf{H} + \mathbf{S}^\top\mathbf{S})\right\| + (1 + \Delta_{\mathcal{M}_D}^S)\|\mathbf{S}\|\right)\|\mathbf{x}^\ell - \mathbf{x}^*\|\square$

### A.4 Proof of Theorem 2

**Theorem 2** (Convergence of NPN Regularization). *Consider the iterations of the PnP-NPN algorithm defined in (4) for $\ell = 1, \ldots, L$. Assume that the estimation error of the trained network $\mathrm{G}^*$ satisfies Assumption 1, and that both $\mathbf{S}$ and $\mathbf{H}$ satisfy the Restricted Isometry Property (RIP) over the manifold $\mathcal{M}_D$ with constants $\Delta_{\mathcal{M}_D}^S, \Delta_{\mathcal{M}_D}^H \in [0,1)$. Further assume that the residual term $\mathrm{N}(\mathbf{H}\mathbf{x}^*)$ is $K$-Lipschitz continuous, thus $\|\mathrm{N}(\mathbf{H}\mathbf{x}^*)\| \leq K(1 + \Delta_{\mathcal{M}_D}^H)\|\mathbf{x}^*\|$. Then, after $\ell$ iterations, the NPN regularization term satisfies the following upper bound:*

$$\|\mathrm{G}(\mathbf{y}) - \mathbf{S}\tilde{\mathbf{x}}^{\ell+1}\| \leq C_1\|\mathbf{x}^* - \mathbf{x}^\ell\| + K\|\mathbf{x}^*\|(1 + \Delta_{\mathcal{M}_D}^H) + C_2\|\mathbf{x}^\ell - \mathbf{x}^{\ell+1}\|, \qquad (10)$$

*where* $C_1 = \left(\frac{1}{2\alpha} + (1 + \Delta_{\mathcal{M}_D}^S)^2 + K(1 + \Delta_{\mathcal{M}_D}^H)(1 + \Delta_{\mathcal{M}_D}^S)\|\mathbf{x}^*\|\right),$ $C_2 = \left((1 + \Delta_{\mathcal{M}_D}^S) + \frac{1}{\sqrt{2\alpha}}\right).$

*Proof:* We begin by recalling the definition of the regularization function:

$$\phi(\mathbf{x}) = \|\mathbf{S}\mathbf{x} - \mathbf{S}\mathbf{x}^* - \mathrm{N}(\mathbf{H}\mathbf{x}^*)\|.$$

From the iteration difference, we have:

$$\phi(\mathbf{x}^{\ell+1}) - \phi(\mathbf{x}^\ell) = \|\mathbf{S}(\mathbf{x}^{\ell+1} - \mathbf{x}^\ell)\|^2 + 2\langle \mathbf{x}^{\ell+1} - \mathbf{x}^\ell, \mathbf{S}^\top(\mathbf{S}\mathbf{x}^\ell - \mathbf{S}\mathbf{x}^* - \mathrm{N}(\mathbf{H}\mathbf{x}^*))\rangle. \qquad (11)$$

Define the intermediate step:

$$\mathbf{q}^\ell = \mathbf{x}^\ell - \alpha\,\mathbf{S}^\top(\mathbf{S}\mathbf{x}^\ell - \mathbf{S}\mathbf{x}^* - \mathrm{N}(\mathbf{H}\mathbf{x}^*)), \quad \alpha > 0.$$

Optimality implies:

$$\|\mathbf{x}^{\ell+1} - \mathbf{q}^\ell\| \leq \|\mathbf{x}^* - \mathbf{q}^\ell\|.$$

Substituting the definition of $\mathbf{q}^\ell$ and rearranging gives:

$$\|\mathbf{x}^{\ell+1} - \mathbf{x}^\ell + \alpha\,\mathbf{S}^\top(\mathbf{S}\mathbf{x}^\ell - \mathbf{S}\mathbf{x}^* - \mathrm{N}(\mathbf{H}\mathbf{x}^*))\|^2 \leq \|\mathbf{x}^* - \mathbf{x}^\ell + \alpha\,\mathbf{S}^\top(\mathbf{S}\mathbf{x}^\ell - \mathbf{S}\mathbf{x}^* - \mathrm{N}(\mathbf{H}\mathbf{x}^*))\|^2.$$

Expanding both sides and reorganizing terms, we obtain:

$$2\alpha\langle \mathbf{x}^{\ell+1} - \mathbf{x}^\ell, \mathbf{S}^\top(\mathbf{S}\mathbf{x}^\ell - \mathbf{S}\mathbf{x}^* - \mathrm{N}(\mathbf{H}\mathbf{x}^*))\rangle \leq \|\mathbf{x}^* - \mathbf{x}^\ell\|^2 - \|\mathbf{x}^{\ell+1} - \mathbf{x}^\ell\|^2$$
$$+ 2\alpha\langle \mathbf{x}^* - \mathbf{x}^\ell, \mathbf{S}^\top(\mathbf{S}\mathbf{x}^\ell - \mathbf{S}\mathbf{x}^* - \mathrm{N}(\mathbf{H}\mathbf{x}^*))\rangle.$$

Dividing by $2\alpha$ and substituting back into (11) yields:

$$\phi(\mathbf{x}^{\ell+1}) - \phi(\mathbf{x}^\ell) \leq \|\mathbf{S}(\mathbf{x}^{\ell+1} - \mathbf{x}^\ell)\|^2 + \frac{1}{2\alpha}\|\mathbf{x}^* - \mathbf{x}^\ell\|^2 - \frac{1}{2\alpha}\|\mathbf{x}^{\ell+1} - \mathbf{x}^\ell\|^2$$
$$+ \langle \mathbf{x}^* - \mathbf{x}^\ell, \mathbf{S}^\top(\mathbf{S}\mathbf{x}^\ell - \mathbf{S}\mathbf{x}^* - \mathrm{N}(\mathbf{H}\mathbf{x}^*))\rangle. \qquad (12)$$

Next, applying Assumptions 2 and 3, we bound:

$$\langle \mathbf{x}^* - \mathbf{x}^\ell, \mathbf{S}^\top(\mathbf{S}\mathbf{x}^\ell - \mathbf{S}\mathbf{x}^* - \mathrm{N}(\mathbf{H}\mathbf{x}^*))\rangle$$
$$\leq \|\mathbf{S}(\mathbf{x}^\ell - \mathbf{x}^*)\|^2 + K(1 + \Delta_{\mathcal{M}_D}^H)(1 + \Delta_{\mathcal{M}_D}^S)\|\mathbf{x}^*\|\|\mathbf{x}^\ell - \mathbf{x}^*\|.$$

Substituting this back into (12), we get:

$$\phi(\mathbf{x}^{\ell+1}) - \phi(\mathbf{x}^\ell) \leq \|\mathbf{S}(\mathbf{x}^{\ell+1} - \mathbf{x}^\ell)\|^2 - \frac{1}{2\alpha}\|\mathbf{x}^{\ell+1} - \mathbf{x}^\ell\|^2$$
$$+ \frac{1}{2\alpha}\|\mathbf{x}^* - \mathbf{x}^\ell\|^2 + \|\mathbf{S}(\mathbf{x}^\ell - \mathbf{x}^*)\|^2$$
$$+ K(1 + \Delta_{\mathcal{M}_D}^H)(1 + \Delta_{\mathcal{M}_D}^S)\|\mathbf{x}^*\|\|\mathbf{x}^\ell - \mathbf{x}^*\|. \qquad (13)$$

Using the triangle inequality and the RIP property again, we simplify:

$$\phi(\mathbf{x}^\ell) \leq (1 + \Delta_{\mathcal{M}_D}^S)\|\mathbf{x}^\ell - \mathbf{x}^*\| + K(1 + \Delta_{\mathcal{M}_D}^H)\|\mathbf{x}^*\|.$$

Hence, regrouping terms, we achieve a final compact form analogous to the desired inequality:

$$\|G(\mathbf{y}) - \mathbf{S}\mathbf{x}^{\ell+1}\| = \phi(\mathbf{x}^{\ell+1})$$

$$\leq \left(\frac{1}{2\alpha} + (1 + \Delta^S_{\mathcal{M}_D})^2 + K(1 + \Delta^H_{\mathcal{M}_D})(1 + \Delta^S_{\mathcal{M}_D})\|\mathbf{x}^*\|\right)\|\mathbf{x}^* - \mathbf{x}^\ell\|$$

$$+ K(1 + \Delta^H_{\mathcal{M}_D})\|\mathbf{x}^*\| + \left((1 + \Delta^S_{\mathcal{M}_D}) + \frac{1}{\sqrt{2\alpha}}\right)\|\mathbf{x}^{\ell+1} - \mathbf{x}^\ell\|. \square \quad (14)$$

### A.5 Neural network details

**SPC**: For the single-pixel camera experiments, we employ a ConvNeXt-inspired backbone [33]. The network has five ConvNeXt blocks, each comprising two successive $3 \times 3$ convolutions with ReLU activations. A final $3 \times 3$ convolutional layer projects the 128 features back to one channel. The output feature map is flattened and passed through a linear module to match the measurement dimensionality. We use cosine-based positional encoding with a dimension of 256, and set the number of blocks to 5 and the base channel width to 128.

**MRI and Deblurring**: To train G for the MRI and Deblurring experiments, we use a U-Net architecture [42] with three downscaling and three upscaling modules. Each module consists of two consecutive Conv $\rightarrow$ ReLU blocks. The downscaling path uses max pooling for spatial reduction, starting with 128 filters and increasing up to 1,024 at the bottleneck. The upscaling path performs nearest-neighbor interpolation before each module, progressively reducing the number of filters. A final 2D convolutional layer without activation produces the output. Skip connections link corresponding layers in the encoder and decoder.

### A.6 Plug and Play algorithms

This work uses the PnP-FISTA, its unrolled version, and RED-FISTA algorithms to validate the proposed approach. Below is shown algorithms for PnP and RED formulation and their NPN-regularized counterparts. For unrolling FISTA, the only change is that the denoiser $D_\sigma$ is changed to a trainable deep neural network that is optimized in an end-to-end manner.

---

**Algorithm 2** PnP-FISTA

**Require:** $L, \mathbf{H}, \mathbf{y}, \alpha$
1: $\mathbf{x}^0 = \mathbf{z}^1 = \mathbf{0}$, $t = 1$
2: **for** $\ell = 1, \ldots, L$ **do**
3: $\quad \mathbf{x}^\ell \leftarrow \mathbf{z}^\ell - \alpha \mathbf{H}^\top(\mathbf{H}\mathbf{z}^\ell - \mathbf{y})$
4: $\quad \mathbf{x}^\ell \leftarrow D_\sigma(\mathbf{x}^\ell)$
5: $\quad t' = t$
6: $\quad t = \frac{1+\sqrt{1+4(t')^2}}{2}$
7: $\quad \mathbf{z}^{\ell+1} \leftarrow \mathbf{x}^\ell + \frac{t'-1}{t}(\mathbf{x}^\ell - \mathbf{x}^{\ell-1})$
8: **end for**
9: **return** $\mathbf{x}^\ell$

---

**Algorithm 3** NPN-PnP-FISTA

**Require:** $L, \mathbf{H}, \mathbf{S}, \mathbf{y}, G^*, \alpha, \gamma$
1: $\mathbf{x}^0 = \mathbf{z}^1 = \mathbf{0}$, $t = 1$
2: **for** $\ell = 1, \ldots, L$ **do**
3: $\quad \mathbf{x}^\ell \leftarrow \mathbf{z}^\ell - \alpha \left(\mathbf{H}^\top(\mathbf{H}\mathbf{z}^\ell - \mathbf{y}) + \gamma\mathbf{S}^\top(\mathbf{S}\mathbf{z}^\ell - G^*(\mathbf{y}))\right)$
4: $\quad \mathbf{x}^\ell \leftarrow D_\sigma(\mathbf{x}^\ell)$
5: $\quad t' = t$
6: $\quad t = \frac{1+\sqrt{1+4(t')^2}}{2}$
7: $\quad \mathbf{z}^{\ell+1} \leftarrow \mathbf{x}^\ell + \frac{t'-1}{t}(\mathbf{x}^\ell - \mathbf{x}^{\ell-1})$
8: **end for**
9: **return** $\mathbf{x}^\ell$

---

**Algorithm 4** RED-FISTA

**Require:** $L, \mathbf{H}, \mathbf{y}, \alpha, \lambda$
1: $\mathbf{x}^0 = \mathbf{z}^1 = \mathbf{0}$, $t = 1$
2: **for** $k = 1, \ldots, K$ **do**
3: $\quad \mathbf{x}^\ell \leftarrow \mathbf{z}^\ell - \alpha\mathbf{H}^\top(\mathbf{H}\mathbf{z}^\ell - \mathbf{y}) - \lambda(\mathbf{z}^\ell - D_\sigma(\mathbf{z}^\ell))$
4: $\quad t' = t$
5: $\quad t = \frac{1+\sqrt{1+4(t')^2}}{2}$
6: $\quad \mathbf{z}^{\ell+1} \leftarrow \mathbf{x}^\ell + \frac{t'-1}{t}(\mathbf{x}^\ell - \mathbf{x}^{\ell-1})$
7: **end for**
8: **return** $\mathbf{x}^\ell$

---

**Algorithm 5** NPN-RED-FISTA

**Require:** $L, \mathbf{H}, \mathbf{S}, \mathbf{y}, G^*, \alpha, \gamma, \lambda$
1: $\mathbf{x}^0 = \mathbf{z}^1 = \mathbf{0}$, $t = 1$
2: **for** $\ell = 1, \ldots, L$ **do**
3: $\quad \mathbf{x}^\ell \leftarrow \mathbf{z}^\ell - \alpha\left(\mathbf{H}^\top(\mathbf{H}\mathbf{z}^\ell - \mathbf{y}) + \gamma\mathbf{S}^\top(\mathbf{S}\mathbf{z}^\ell - G^*(\mathbf{y}))\right) - \lambda(\mathbf{z}^\ell - D_\sigma(\mathbf{z}^\ell))$
4: $\quad t' = t$
5: $\quad t = \frac{1+\sqrt{1+4(t')^2}}{2}$
6: $\quad \mathbf{z}^{\ell+1} \leftarrow \mathbf{x}^\ell + \frac{t'-1}{t}(\mathbf{x}^\ell - \mathbf{x}^{\ell-1})$
7: **end for**
8: **return** $\mathbf{x}^\ell$

## A.7 Deep Image Prior

Deep Image Prior (DIP) [54] is an unsupervised reconstruction framework that leverages untrained neural networks to reconstruct the underlying signal. This approach, since it only compares with the measurements $\mathbf{y}$, its solution may overfit to the noise of the measurements. Therefore, the use of the NPN provides a suitable approach to improve image consistency reconstructions. The optimization problem with the proposed NPN regularization is

$$\mathrm{K}^* = \underset{\mathrm{K}}{\arg\min} \, \|\mathbf{y} - \mathbf{H}\mathrm{K}(\mathbf{z})\|_2^2 + \gamma\|\mathrm{G}^*(\mathbf{y}) - \mathbf{S}\mathrm{K}(\mathbf{z})\|_2^2, \quad \hat{\mathbf{x}} = \mathrm{K}^*(\mathbf{z}), \quad \mathbf{z} \sim \mathcal{N}(0,1), \quad (15)$$

where $\gamma$ is a regularization parameter.

## A.8 Diffusion-based Solvers

Diffusion models (DMs) [47, 22, 49, 50] have recently gained attention due to their exceptional capability in modeling complex image distributions via an iterative noising-denoising process. Conditioning DMs entails guiding their generative reverse diffusion process using measured data to ensure reconstructions align with the measurements. We have integrated our proposed regularization term into two widely adopted diffusion-model frameworks, Diffusion Posterior Sampling (DPS) [12] and DiffPIR [65]. These frameworks serve as canonical diffusion pipelines upon which newer methods build. Our regularization could likewise be incorporated into other approaches, such as latent-space diffusion models [15, 48].

**DPS [12]:** we denote $N$ is the number of reverse diffusion steps, and $i \in 0, \ldots, N-1$ is the reverse-time index; $\mathbf{x}_i \in \mathbb{R}^n$ is the current latent state and $\mathbf{x}_N \sim \mathcal{N}(\mathbf{0}, \mathbf{I})$ is the Gaussian start; $\hat{\mathbf{s}} = \mathbf{s}_\theta(\mathbf{x}_i, i)$ is the score/noise estimate produced by the network with parameters $\theta$; $\hat{\mathbf{x}}_0$ is the network's prediction of the clean sample at step $i$; $\alpha_i \in (0, 1]$ is the per-step retention factor, $\beta_i = 1 - \alpha_i$ is the noise increment, and $\bar{\alpha}_i = \prod_{j=1}^{i} \alpha_j$ is the cumulative product (with $\bar{\alpha}_0 = 1$); $\zeta_i > 0$ is the data-consistency step size and $\tilde{\sigma}_i \geq 0$ is the sampling noise scale at step $i$; $\mathbf{z} \sim \mathcal{N}(\mathbf{0}, \mathbf{I})$ is i.i.d. Gaussian noise; $\mathbf{x}'^{i-1}$ denotes the pre–data-consistency iterate before applying the gradient correction; $\nabla_{\mathbf{x}_i}\|\mathbf{y} - \mathbf{H}\hat{\mathbf{x}}_0\|_2^2$ is the gradient of the quadratic data-fidelity term with respect to $\mathbf{x}_i$ (through the dependence $\hat{\mathbf{x}}_0(\mathbf{x}_i)$);

---

**Algorithm 6** DPS Sampling

**Require:** $N$, $\mathbf{H}$, $\mathbf{y}$, $\{\zeta_i\}_{i=1}^{N}$, $\{\tilde{\sigma}_i\}_{i=1}^{N}$ ▷ step sizes and noise scales
1: $\mathbf{x}_N \sim \mathcal{N}(\mathbf{0}, \mathbf{I})$
2: **for** $i = N-1, \ldots, 0$ **do**
3: $\quad \hat{\mathbf{s}} \leftarrow \mathbf{s}_\theta(\mathbf{x}_i, i)$
4: $\quad \hat{\mathbf{x}}_0 \leftarrow \frac{1}{\sqrt{\bar{\alpha}_i}}(\mathbf{x}_i + (1 - \bar{\alpha}_i)\hat{\mathbf{s}})$
5: $\quad \mathbf{z} \sim \mathcal{N}(\mathbf{0}, \mathbf{I})$
6: $\quad \mathbf{x}'_{i-1} \leftarrow \frac{\sqrt{\alpha_i}(1-\bar{\alpha}_{i-1})}{1-\bar{\alpha}_i} \mathbf{x}_i + \frac{\sqrt{\bar{\alpha}_{i-1}}\beta_i}{1-\bar{\alpha}_i} \hat{\mathbf{x}}_0 + \tilde{\sigma}_i \mathbf{z}$
7: $\quad \mathbf{x}_{i-1} \leftarrow \mathbf{x}'_{i-1} - \zeta_i \nabla_{\mathbf{x}_i}\|\mathbf{y} - \mathbf{H}\hat{\mathbf{x}}_0\|_2^2$
8: **end for**
9: **return** $\hat{\mathbf{x}}_0$

---

**Algorithm 7** NPN-DPS Sampling

**Require:** $N$, $\mathbf{H}$, $\mathbf{y}$, $\{\zeta_i\}_{i=1}^{N}$, $\{\tilde{\sigma}_i\}_{i=1}^{N}$ ▷ step sizes and noise scales
1: $\mathbf{x}_N \sim \mathcal{N}(\mathbf{0}, \mathbf{I})$
2: **for** $i = N-1, \ldots, 0$ **do**
3: $\quad \hat{\mathbf{s}} \leftarrow \mathbf{s}_\theta(\mathbf{x}_i, i)$ ▷ score / noise-prediction net
4: $\quad \hat{\mathbf{x}}_0 \leftarrow \frac{1}{\sqrt{\bar{\alpha}_i}}(\mathbf{x}_i + (1 - \bar{\alpha}_i)\hat{\mathbf{s}})$
5: $\quad \mathbf{z} \sim \mathcal{N}(\mathbf{0}, \mathbf{I})$
6: $\quad \mathbf{x}'_{i-1} \leftarrow \frac{\sqrt{\alpha_i}(1-\bar{\alpha}_{i-1})}{1-\bar{\alpha}_i} \mathbf{x}_i + \frac{\sqrt{\bar{\alpha}_{i-1}}\beta_i}{1-\bar{\alpha}_i} \hat{\mathbf{x}}_0 + \tilde{\sigma}_i \mathbf{z}$
7: $\quad \mathbf{x}_{i-1} \leftarrow \mathbf{x}'_{i-1} - \zeta_i(\nabla_{\mathbf{x}_i}\|\mathbf{y} - \mathbf{H}\hat{\mathbf{x}}_0\|_2^2 + \gamma\|\mathrm{G}^*(\mathbf{y}) - \mathbf{S}\hat{\mathbf{x}}_0\|)$
8: **end for**
9: **return** $\hat{\mathbf{x}}_0$

---

**DiffPIR [65].** $\sigma_n > 0$ denotes the standard deviation of the measurement noise, $\lambda > 0$ is the data–proximal penalty that trades off data fidelity and the denoiser prior inside the subproblem; $\rho_i \triangleq \lambda, \sigma_n^2/\tilde{\sigma}_i^2$ is the iteration-dependent weight used in the proximal objective at step $i$; $\tilde{\mathbf{x}}_0^{(i)}$ is the score-model denoised prediction of the clean sample at step $i$ (before enforcing data consistency); $\hat{\mathbf{x}}_0^{(i)}$ is the solution of the data-proximal subproblem at step $i$ (i.e., the data-consistent refinement of $\tilde{\mathbf{x}}_0^{(i)}$); $\hat{\boldsymbol{\epsilon}} = (1 - \alpha_i)^{-1/2}\left(\mathbf{x}_i - \sqrt{\bar{\alpha}_i}, \hat{\mathbf{x}}_0^{(i)}\right)$ is the effective noise estimate implied by $(\mathbf{x}_i, \hat{\mathbf{x}}_0^{(i)})$; $\boldsymbol{\epsilon}_i \sim \mathcal{N}(\mathbf{0}, \mathbf{I})$ is the fresh Gaussian noise injected at step $i$; $\zeta \in [0, 1]$ mixes deterministic and stochastic updates in the reverse diffusion ($\zeta = 0$ fully deterministic, $\zeta = 1$ fully stochastic).

| **Algorithm 8** DiffPIR Sampling | **Algorithm 9** NPN–DiffPIR Sampling |
|---|---|

**Algorithm 8** DiffPIR Sampling

**Require:** $N$, $\mathbf{H}$, $\mathbf{y}$, $\sigma_n$, $\{\tilde{\sigma}_i\}_{i=1}^N$, $\zeta$, $\lambda$
1: Precompute $\rho_i \leftarrow \lambda \sigma_n^2 / \tilde{\sigma}_i^2$ for $i = 1, \ldots, N$
2: $\mathbf{x}_N \sim \mathcal{N}(\mathbf{0}, \mathbf{I})$
3: **for** $i = N, \ldots, 1$ **do**
4: $\quad \hat{\mathbf{s}} \leftarrow \mathbf{s}_\theta(\mathbf{x}_i, i)$
5: $\quad \tilde{\mathbf{x}}_0^{(i)} \leftarrow \frac{1}{\sqrt{\bar{\alpha}_i}}(\mathbf{x}_i + (1 - \bar{\alpha}_i)\hat{\mathbf{s}})$
6: $\quad \hat{\mathbf{x}}_0^{(i)} \leftarrow \arg\min_{\mathbf{x}} \ \|\mathbf{y} - \mathbf{H}\mathbf{x}\|_2^2 + \rho_i \|\mathbf{x} - \tilde{\mathbf{x}}_0^{(i)}\|_2^2$
7: $\quad \hat{\boldsymbol{\epsilon}} \leftarrow \frac{1}{\sqrt{1-\bar{\alpha}_i}}\left(\mathbf{x}_i - \sqrt{\bar{\alpha}_i}\hat{\mathbf{x}}_0^{(i)}\right)$
8: $\quad \boldsymbol{\epsilon}_i \sim \mathcal{N}(\mathbf{0}, \mathbf{I})$
9: $\quad \mathbf{x}_{i-1} \leftarrow \sqrt{\bar{\alpha}_{i-1}}\hat{\mathbf{x}}_0^{(i)} + \sqrt{1 - \bar{\alpha}_{i-1}}(\sqrt{1-\zeta}\,\hat{\boldsymbol{\epsilon}} + \sqrt{\zeta}\,\boldsymbol{\epsilon}_i)$
10: **end for**
11: **return** $\hat{\mathbf{x}}_0^{(1)}$

**Algorithm 9** NPN–DiffPIR Sampling

**Require:** $N$, $\mathbf{H}$, $\mathbf{y}$, $\sigma_n$, $\{\tilde{\sigma}_i\}_{i=1}^N$, $\zeta$, $\lambda$, $\gamma$, $\mathrm{G}^*$
1: Precompute $\rho_i \leftarrow \lambda \sigma_n^2 / \tilde{\sigma}_i^2$ for $i = 1, \ldots, N$
2: $\mathbf{x}_N \sim \mathcal{N}(\mathbf{0}, \mathbf{I})$
3: **for** $i = N, \ldots, 1$ **do**
4: $\quad \hat{\mathbf{s}} \leftarrow \mathbf{s}_\theta(\mathbf{x}_i, i)$
5: $\quad \tilde{\mathbf{x}}_0^{(i)} \leftarrow \frac{1}{\sqrt{\bar{\alpha}_i}}(\mathbf{x}_i + (1 - \bar{\alpha}_i)\hat{\mathbf{s}})$
6: $\quad \hat{\mathbf{x}}_0^{(i)} \leftarrow \arg\min_{\mathbf{x}} \ \|\mathbf{y} - \mathbf{H}\mathbf{x}\|_2^2 + \rho_i \|\mathbf{x} - \tilde{\mathbf{x}}_0^{(i)}\|_2^2 + \gamma \|\mathrm{G}^*(\mathbf{y}) - \mathbf{S}\mathbf{x}\|_2^2$
7: $\quad \hat{\boldsymbol{\epsilon}} \leftarrow \frac{1}{\sqrt{1-\bar{\alpha}_i}}\left(\mathbf{x}_i - \sqrt{\bar{\alpha}_i}\hat{\mathbf{x}}_0^{(i)}\right)$
8: $\quad \boldsymbol{\epsilon}_i \sim \mathcal{N}(\mathbf{0}, \mathbf{I})$
9: $\quad \mathbf{x}_{i-1} \leftarrow \sqrt{\bar{\alpha}_{i-1}}\hat{\mathbf{x}}_0^{(i)} + \sqrt{1 - \bar{\alpha}_{i-1}}(\sqrt{1-\zeta}\,\hat{\boldsymbol{\epsilon}} + \sqrt{\zeta}\,\boldsymbol{\epsilon}_i)$
10: **end for**
11: **return** $\hat{\mathbf{x}}_0^{(1)}$

## A.9 Implementation details on NSN-based methods

Recall the NSN-based models used in comparison:

1. Deep null space network (DNSN)[45]: $\hat{\mathbf{x}} = \mathbf{H}^\dagger \mathbf{y} + (\mathbf{I} - \mathbf{H}^\dagger \mathbf{H})\mathrm{R}(\mathbf{H}^\dagger \mathbf{y})$.

2. Deep decomposition network cascade (DDN-C) [6]: $\hat{\mathbf{x}} = \mathbf{H}^\dagger \mathbf{y} + \mathrm{P}_r(\mathrm{R}_r(\mathbf{H}^\dagger \mathbf{y})) + \mathrm{P}_n(\mathrm{R}_n(\mathbf{H}^\dagger \mathbf{y} + \mathrm{P}_r(\mathrm{R}_r(\mathbf{H}^\dagger \mathbf{y}))))$.

3. Deep decomposition network independent (DDN-I) [6]: $\hat{\mathbf{x}} = \mathbf{H}^\dagger \mathbf{y} + \mathrm{P}_r(\mathrm{R}_r(\mathbf{H}^\dagger \mathbf{y})) + \mathrm{P}_n(\mathrm{R}_n(\mathbf{H}^\dagger \mathbf{y}))$

We used the source code [2] of [6] to implement the models $\mathrm{R}, \mathrm{R}_r$ and $\mathrm{R}_n$.

- Network $\mathrm{R}$ and $\mathrm{R}_n$: is a lightweight version of the U-Net architecture for image segmentation. It features an encoder-decoder structure with skip connections. The encoder consists of five convolutional blocks, each followed by max-pooling to extract features. The decoder upsamples using transposed convolutions and refines the features with additional convolutions, leveraging skip connections to combine low- and high-level features. The final output is produced through a $1 \times 1$ convolution, reducing the output to the desired number of channels. This compact design makes it efficient for tasks with limited resources.

- Network $\mathrm{R}_r$: It consists of a series of convolutional layers arranged in a sequential block structure, where each block performs a transformation of the input image through convolutional operations, followed by activation functions, such as ReLU, and normalization techniques like batch normalization. The model starts with an input image, and through several layers of convolutions, the features of the image are progressively refined. Each layer extracts relevant features, while ReLU activations introduce non-linearity to improve the model's capacity to capture complex patterns. Batch normalization layers are added to stabilize training and speed up convergence by normalizing the output of each convolutional layer. The final output layer of the network reconstructs the denoised image. The architecture is designed with a depth of 17 layers.

All models were trained for 100 epochs using the Adam optimizer with a learning rate of 1e-3 using a mean-squared-error loss function.

## A.10 Super resolution experiments

For this scenario, we use the Places365 dataset We set an SR factor $SRF = \sqrt{\frac{n}{m}} = 4$; downsampling was performed with bilinear interpolation. The forward operator is modeled as $\mathbf{H} = \mathbf{DB} \in \mathbb{R}^{m \times n}$,

---

[2]https://github.com/edongdongchen/DDN

| Metric | PnP-FISTA Baseline | PnP-FISTA NPN | Estimation of $\mathbf{Sx}$ |
|--------|--------------------|--------------------|------------------|
| PSNR | 22.01 | 23.75 | **24.95** |
| SSIM | 0.562 | 0.692 | **0.697** |

Table 4: Quantitative comparison of PnP-FISTA and PnP-FISTA NPN with the estimation of $\mathbf{Sx}$.

where $\mathbf{B} \in \mathbb{R}^{n \times n}$ is a structured Toeplitz matrix implementing Gaussian blur with bandwidth $\sigma = 2.0$ and $\mathbf{D} \in \mathbb{R}^{m \times n}$ is the downsampling matrix.

To recover the information removed by $\mathbf{H}$, we define the null-space projection operator, similar to the deblurring case, $\mathbf{S}[i, i + j] = 1 - \mathbf{h}[j], \quad i = 1, \ldots, m, \quad j = 1, \ldots, n$, which captures the high-frequency details of the signal. We use the PnP-FISTA algorithm for evaluation with 60 iterations using $\alpha = 0.5$ and $\gamma = 0.1$. In the following table, we show the reconstruction performance with baseline PnP-FISTA and NPN PnP-FISTA, showing an improvement of 1.74 dB in PSNR and 0.13 in SSIM. Additionally, we show the network estimation of the null-space $\mathbf{Sx}$ metrics, showing good estimation of the high-frequency details from the low-resolution image.

## A.11   Additional Experiments

In Tables 5 and 6, we provide additional results of MRI and deblurring, respectively, using PnP, RED, and sparsity priors for acceleration factors (AFs) of 4, 8, 12 (with 5 dB SNR measurement noise) and $\sigma$ of 2, 6, 10. These results validate the robustness of NPN regularization under different numbers of measurements.

| AF | Method | Prior | | |
|----|--------|-----|----------|-----|
|    |        | PnP | Sparsity | RED |
| 4 | Baseline | 30.91 | 30.05 | 29.17 |
|   | NPN | 31.11 | 31.04 | 31.46 |
| 8 | Baseline | 27.25 | 26.72 | 27.59 |
|   | NPN | 29.88 | 30.02 | 28.84 |
| 12 | Baseline | 26.35 | 25.68 | 27.35 |
|    | NPN | 29.06 | 29.24 | 28.13 |

Table 5: MRI: PSNR (dB) for PnP vs. PnP–NPN under different priors and AFs.

| $\sigma$ | Method | Prior | | |
|----------|--------|-----|----------|-----|
|          |        | PnP | Sparsity | RED |
| 2.0 | PnP | 30.78 | 29.27 | 32.84 |
|     | NPN | 31.77 | 31.42 | 33.67 |
| 5.0 | PnP | 24.47 | 23.17 | 24.79 |
|     | NPN | 25.72 | 25.32 | 25.34 |
| 10.0 | PnP | 20.49 | 19.61 | 20.48 |
|      | NPN | 20.81 | 20.60 | 20.54 |

Table 6: Deblurring: PSNR (dB) for PnP vs. PnP-NPN under different priors and noise levels.

In Figures 7, 8, and 9, the effect of the parameter $\gamma$ on the quality of the recovery is shown, for MRI, Deblurring, and SPC, respectively.

Fig. 10 shows the results in terms of convergence of the PnP and PnP-NPN for SPC with $m/n = 0.1$. The color of the line indicates the type of projection matrix used, orthonormal by QR (red) or designed by Eq. 3 (blue). The color shade indicates the percentage of the low-dimensional subspace $p/n$, ranging from 0.1 to 0.9. The best case is with $p/n = 0.1$ and $\mathbf{S}$ designed, which was the one used in the experiments of the main paper.

Additionally, we implemented several state-of-the-art denoisers into the PnP framework. We also change the FISTA algorithm used in the paper to an alternating direction method of multipliers (ADMM) formulation. This algorithm splits the optimization problem into two subproblems: the data fidelity and the prior. We employed several state-of-the-art denoisers such as Restormer [60], DnCNN [63], DnCNN-Lipschitz [43], and DRUNet [62]. We validated the image deblurring application with a Gaussian kernel bandwidth $\sigma = 2.0$. We used 200 iterations of the PnP-ADMM algorithm, the step size $\alpha = 0.5$, and the value of $\gamma = 0.7$. In Table 1 are shown the obtained results. Here, the proposed regularization function consistently improves the baseline PnP-ADMM algorithm.

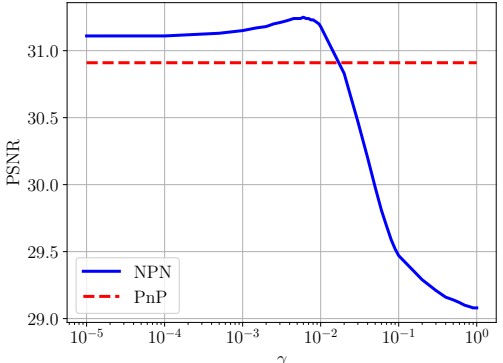

Figure 7: Effect of $\gamma$ in NPN on PSNR (dB) for MRI reconstruction, with $\alpha = 1 \times 10^{-4}$. The maximum PSNR of 33.67 dB is achieved when $\gamma = 6 \times 10^{-3}$.

Figure 8: Effect of $\gamma$ on PSNR (dB) in deblurring reconstruction, with $\alpha = 0.5 \times 10^{-4}$. The maximum PSNR of 31.25 dB is achieved when $\gamma = 0.263$.

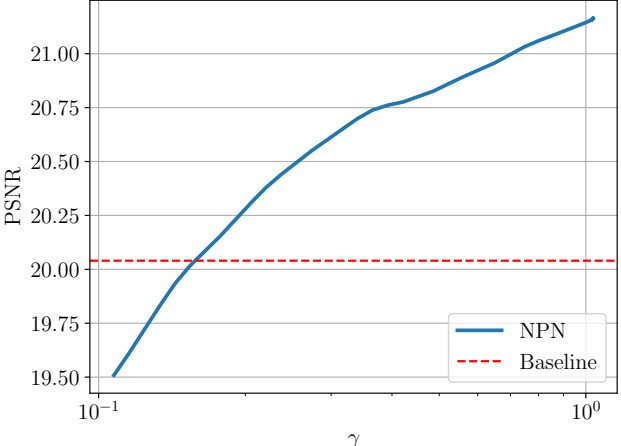

Figure 9: Effect of $\gamma$ on PSNR (dB) in SPC reconstruction, with $\alpha = 8 \times 10^{-4}$. The maximum PSNR of 21.17 dB is achieved when $\gamma = 1.04$.

| Method | Restormer | DnCNN | DnCNN-Lipschitz | DRUNet | Sparsity Prior |
|---|---|---|---|---|---|
| Baseline | 29.86 | 29.55 | 30.36 | 29.68 | 28.75 |
| NPN | **32.62** | 32.12 | 32.35 | 32.07 | 29.75 |

Table 7: Comparison of PnP-ADMM method for image deblurring with $\sigma = 2.0$ using different denoisers. The best result is highlighted in **bold teal** , and the second-best in orange underline .

## A.12 Analysis on the selction of $p$ and data adaptation

The selection of the size of $\mathbf{S}$ is important for the optimization (3) as the bigger value of $p$, the more challenging the projection estimation becomes. We show the estimation error in the following table, showing that the error increases by increasing the number of rows of $\mathbf{S}$.

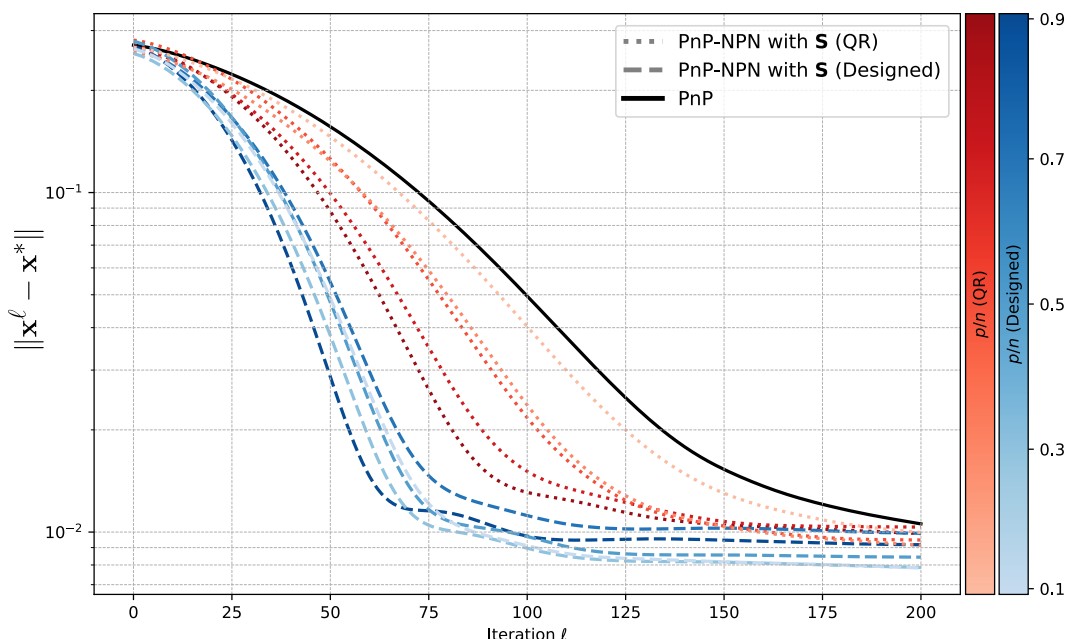

Figure 10: Effect of the low-dimensional subspace dimension $p/n$ and projection matrix $\mathbf{S}$ design on signal convergence in SPC reconstruction, with $\alpha = 8 \times 10^{-4}$.

Table 8: Projection error for different values of $p/n$.

| $p/n$ | $\frac{\|\mathbf{Sx}^* - \mathbf{G(y)}\|}{\|\mathbf{Sx}^*\|}$ |
|---|---|
| 0.1 | 0.1031 |
| 0.3 | 0.1544 |
| 0.5 | 0.1821 |
| 0.7 | 0.2566 |
| 0.9 | 0.2305 |

Additionally, to illustrate the effect of the data-driven adaptability of $\mathbf{S}$, we clarify the following. In compressed sensing, to span the null-space, we design a matrix orthogonal to $\mathbf{H}$ via the QR decomposition, and via optimization using the regularization in Eq. (3), which promotes that the rows of $\mathbf{S}$ lie in the null-space of $\mathbf{H}$. To analyze how the data distribution affects the design of $\mathbf{S}$, we optimize four matrices, one for each dataset (MNIST, FashionMNIST, CIFAR10, and CelebA), cross-validating the data-driven invertibility loss $\mathbb{E}_\mathbf{x}\|\mathbf{x} - \mathbf{S}^\dagger\mathbf{Sx}\|_2^2$. The results show that the QR-initialized matrix provides good invertibility across all datasets, and that optimizing $\mathbf{S}$ via Eq. (3) on each dataset further improves it. Thus, the data distribution can enhance the data-driven invertibility of an already orthogonal matrix. Testing $\mathbf{S}$ on in-distribution samples yields the **best** invertibility (**bold**), while diverse training data (e.g. CIFAR10) gives the second-best (underlined), suggesting a single $\mathbf{S}$ could generalize across datasets.

| Test/**S** design | $\mathcal{QR}(\mathbf{H})$ Alg 1 | **MNIST** | **FashionMNIST** | **CIFAR10** | **CelebA** |
|---|---|---|---|---|---|
| **MNIST** | $9.28 \times 10^{-5}$ | $\mathbf{2.97 \times 10^{-5}}$ | $\underline{4.99 \times 10^{-5}}$ | $5.29 \times 10^{-5}$ | $5.56 \times 10^{-5}$ |
| **FashionMNIST** | $1.76 \times 10^{-4}$ | $7.07 \times 10^{-5}$ | $\mathbf{3.11 \times 10^{-5}}$ | $\underline{5.05 \times 10^{-5}}$ | $5.62 \times 10^{-5}$ |
| **CIFAR10** | $2.62 \times 10^{-4}$ | $1.67 \times 10^{-4}$ | $8.95 \times 10^{-5}$ | $\mathbf{4.01 \times 10^{-5}}$ | $\underline{4.52 \times 10^{-5}}$ |
| **CelebA** | $2.39 \times 10^{-4}$ | $1.55 \times 10^{-4}$ | $9.09 \times 10^{-5}$ | $\underline{4.68 \times 10^{-5}}$ | $\mathbf{4.15 \times 10^{-5}}$ |

Table 9: Cross dataset validation of data invertibility metric $\mathbb{E}_\mathbf{x}\|\mathbf{x} - \mathbf{S}^\dagger\mathbf{Sx}\|_2^2$

## A.13 Limitations on the selection of S

The method works when it is possible to find a non-linear correlation between null space (NS) components $\mathbf{Sx}$ and measurements $\mathbf{Hx}$, leveraging a dataset with triplets $(\mathbf{x}_i, \mathbf{Hx}_i, \mathbf{Sx}_i)$. We promote this correlation by solving Eq. (3) that balances the two terms: i) $\|\mathbf{Sx} - \mathrm{G}(\mathbf{Hx})\|$ encourages a non-linear correlation between $\mathbf{Sx}$ and $\mathbf{Hx}$. ii) $\|\mathbf{I} - \mathbf{A}^\top \mathbf{A}\|$ where $\mathbf{A} = [\mathbf{H}^\top, \mathbf{S}^\top]^\top$ promotes orthogonality and ensures that $\mathbf{S}$ samples components from the NS of $\mathbf{H}$.

There are scenarios where the non-linear correlations are easier to achieve, for instance, in deblurring or SR, due to the low-pass filters associated with these tasks being non-ideal, which leads to close frequency bands between the high-pass components selected by $\mathbf{S}$ and the Gaussian-like low-pass components sampled by $\mathbf{H}$. In other scenarios, such as MRI or CT, the non-linear correlations between $\mathbf{Sx}$ and $\mathbf{Hx}$ are more challenging to achieve due to the orthogonality of their respective rows. But, the method still works by designing $\mathbf{S}$ such that it leverages structural similarities of $\mathbf{H}$, such as adjacent frequencies in MRI or neighboring angles in CT.

To illustrate the effectiveness of finding non-linear correlations between $\mathbf{Sx}$ and $\mathbf{Hx}$, we set a challenging scenario in an MRI task with $\mathbf{H}$ sampling only random low frequencies using a 1D Cartesian mask with an acceleration factor (AF) of 12. We then apply an FFTSHIFT operation to this mask to produce a high-frequency sampling pattern, denoted by $\mathbf{S}_1$, spatially distant from the frequency support of $\mathbf{H}$ but maintaining the same AF. We refer to this configuration as the well-separated sampling. In this case, the method fails to recover non-trivial solutions, $\|\mathrm{G}^*(\mathbf{y})\| = 0.0108$, and a high relative error of 1.0.

Then, we configure a scenario where the non-linear correlations are easier to find, using the same $\mathbf{H}$ described before, we set $\mathbf{S}_2 = \mathbf{I} - \mathbf{H}$ that selects frequencies adjacent to those sampled by $\mathbf{H}$. This adjacent sampling configuration corresponds precisely to the setup employed in the MRI experiments presented in the main manuscript. See Figure 11. It exhibits increased spectral (non-linear) correlation, resulting in a small relative error of 0.325, and a norm $\|\mathrm{G}^*(\mathbf{y})\|$ close to the norm of the ground-truth signal $\|\mathbf{Sx}\|$.

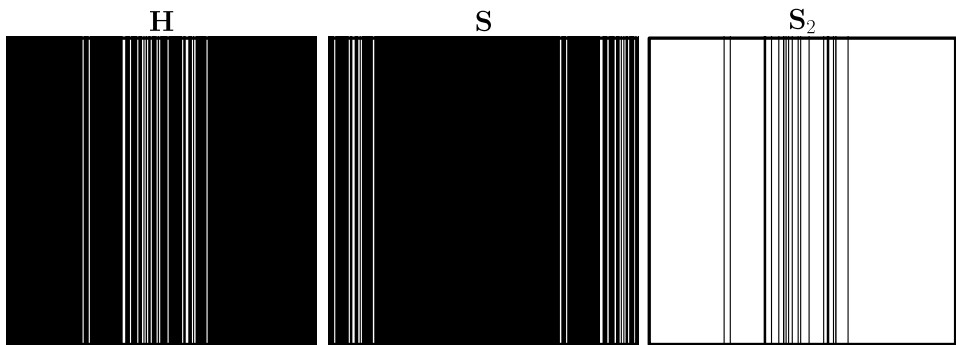

Figure 11: MRI masks

| Sampling Type | $\dfrac{\|\mathbf{Sx} - \mathrm{G}(\mathbf{y})\|}{\|\mathbf{Sx}\|}$ | $\|\mathrm{G}(\mathbf{y})\|$ | $\|\mathbf{Sx}\|$ | PSNR($\mathrm{G}(\mathbf{y}), \mathbf{Sx}$) [dB] |
|---|---|---|---|---|
| Disjoint Sampling | 1.00 | 0.0197 | 1.72 | 30.7 |
| Adjacent Sampling | 0.40 | 98.7 | 112 | 59.5 |

Table 10: Comparison of reconstruction metrics under different sampling strategies.

## A.14 Comparison with reconstruction model

To create a fair comparison with the proposed method, we consider the network $\mathrm{W}^* : \mathbb{R}^m \to \mathbb{R}^n$, which has the same number of parameters as $\mathrm{G}$ but is trained to reconstruct directly the underlying signal i.e., $\mathrm{W}^* = \arg\min_{\mathrm{W}} \mathbb{E}_{\mathbf{x},\mathbf{y}} \|\mathbf{x} - \mathrm{W}(\mathbf{y})\|$. The network was trained with the same number of epochs as $\mathrm{G}$ (300 epochs), using the AdamW optimizer with a learning rate of $1e^{-4}$. We evaluate two aspects here: i) the reconstruction performance of the network $\mathrm{W}$ and ii) incorporating into the PnP-ADMM the regularization but in the form of $\phi_{\mathrm{W}}(\mathbf{x}) = \|\mathbf{Sx} - \mathbf{SW}^*(\mathbf{y})\|_2$. We trained the

network W with the Places365 dataset. The PnP-ADMM was set with 100 iterations, using the DnCNN-Lipchitz denoiser. We set $\alpha = 0.3$ and $\gamma = 0.2$. In the following table, we report the obtained results. The results show that the proposed approach overall improves the baseline, the reconstruction of the model W, and the regularization based on this model.

| Metric | Base PnP-ADMM | NPN PnP-ADMM | PnP-ADMM with $\phi_W$ | $W^*(\mathbf{y})$ |
|---|---|---|---|---|
| PSNR (dB) | 22.30 | 23.87 | 21.20 | 19.71 |
| SSIM | 0.586 | 0.678 | 0.534 | 0.490 |

Table 11: Comparison of PSNR (dB) and SSIM metrics

In the tables below, we show the training and execution time in seconds (s) and minutes (min), respectively. These results are obtained for a batch size of 100 images with a resolution of $128 \times 128$ and 100 iterations of the PnP-ADMM. W (reconstruction network from the measurements $\mathbf{y}$) and G are trained for 300 epochs. The training times of W and G are very similar due to having the same number of parameters. Despite a modest increase of 0.20 seconds over the baseline PnP-ADMM, NPN achieves a 1.57 dB PSNR gain. Moreover, compared to PnP-ADMM with $\phi_W$, NPN is 0.28 seconds faster while delivering a 2.67 dB improvement in PSNR. These results show that the NPN regularization yields substantial quality gains with minimal or favorable time trade-offs.

| Method | Training time (min) |
|---|---|
| G($\mathbf{y}$) | 385 |
| W($\mathbf{y}$) | 390 |

Table 12: Training time for the networks W and G (NPN).

| Method | Execution time (s) |
|---|---|
| Base PnP-ADMM | **12.48** |
| NPN PnP-ADMM | *12.68* |
| PnP-ADMM with $\phi_W$ | 12.96 |

Table 13: Execution times for PnP with network W and G (NPN).

