# OpenReview forum: "NPN: Non-Linear Projections of the Null-Space for Imaging Inverse Problems"
_NeurIPS.cc/2025/Conference — NeurIPS 2025 poster_

### Official Review · Reviewer_PAt7 · 2025-06-04

**Clarity:** 4
**Significance:** 2
**Originality:** 3
**Rating:** 4
**Confidence:** 4

**Summary:**

The paper introduces a method for solving imaging inverse problems by leveraging the structure of the data in the null space of the measurement operator. The authors propose a novel regularization approach that uses a neural network trained to estimate the null space component from measurements in a supervised manner. Reconstruction is then performed via a Plug-and-Play (PnP) scheme that uses the gradient of the learned regularization.

**Questions:**

### Suggestions

**Theory**

- Clearly state whether the convergence result of Theorem 1 applies to the infinite sequence or only a finite number of iterations.

- Provide practical guidance on how to choose step size $\alpha$ and offer examples or justification for when the operator $S$ satisfies the required conditions (e.g., low spectral norm, low RIC $\Delta^S_{M_D}$).

- Include theoretical or empirical analysis to demonstrate that the convergence improvement zone is non-empty and reachable in practice.

**Experiments**

- Improve baseline comparisons and include more datasets and problems to give more information to the reader.

- Provide results showing the effect of different PnP denoisers, step sizes, and regularization parameters ($\lambda_1$, $\lambda_2$) and discuss how sensitive the method is to these hyperparameters in practice.

**Ethical Concerns:**

["NO or VERY MINOR ethics concerns only"]

**Final Justification:**

I upgraded my rating after the rebuttal and discussion with the authors, which helped clarify the practical interest of the theory. I recommend acceptance of the paper, though the authors should include all the details given in the discussion to the paper to make it clearer.

**Limitations:**

The authors discussed limitations.

**Quality:**

2

**Strengths And Weaknesses:**

### Strengths

- The proposed method leverages the structure of the null space, a promising and active direction in inverse problems.

- The paper is clearly written and well-organized, making the technical content accessible.

### Weaknesses

- The theoretical analysis and its practical relevance are somewhat unclear:
    * In Theorem 1, the authors claim that the algorithm converges to a fixed point, but the convergence analysis is limited to a possibly finite number of iterations.
    * Lines 239–251 offer informal justification for the assumptions required for convergence. However, the paper lacks both practical and theoretical evidence that these assumptions hold in realistic settings. For example, what would be an appropriate step size $\alpha$? What kind of operator satisfies the condition that the spectral norm of $S$ and its RIC $\Delta^S_{M_D}$ are sufficiently small?
    * Since the "convergence" behavior only holds within the "convergence improvement zone," it would be helpful to know whether this zone is non-empty and practically reachable. Can the authors provide empirical or theoretical evidence that iterates typically enter this region?

- The experimental section could be significantly strengthened:
    * The chosen baseline appears weak. More recent and competitive methods are readily available (see https://deepinv.github.io/deepinv/ for a list of algorithms and networks).
    * It would benefit from additional datasets and a wider variety of inverse problems and/or parameterizations.
    * The sensitivity of the method to key hyperparameters—such as step size, PnP denoiser choice, and regularization weights $\lambda_1$ and $\lambda_2$—is not thoroughly explored. From a practical standpoint, it would be valuable to understand how robust the method is to these choices.

---

> ### Author Rebuttal · Authors · 2025-07-31
>
> We appreciate the reviewer’s constructive feedback, which has helped improve the paper. We also thank the reviewer for noting that the proposed method is “a promising and active direction in inverse problems”. Below, we address each concern in detail.
>
> # Weaknesses
> ---
> ## W1. Clarity on Theorem 1
> We thank the reviewer for the insightful comment and agree that the fixed-point convergence claim in the original statement of Theorem 1 may be misleading. Our main contribution in this result is not to prove convergence to a unique fixed point, but rather to establish a linear residual decay rate within a bounded number of iterations, defined as the *convergence improvement zone* (CIZ)  Specifically, Theorem 1 shows that the error $\Vert\mathbf{x}^{\ell+1} - \mathbf{x}^\ast\Vert_2^2$
> decreases at a linear rate:
> $$\rho \triangleq (1+\delta) \left( \left\Vert \mathbf{I} - \alpha (\mathbf{H}^{\top} \mathbf{H} + \mathbf{S}^{\top} \mathbf{S}) \right\Vert_2^2 + (1+\Delta_{\mathcal{M}_D}^S) \Vert \mathbf{S}\Vert_2^2\right),$$
> with respect to the previous iteration error $\Vert\mathbf{x}^{\ell} - \mathbf{x}^\ast\Vert$. Here, the dominant terms controlling the decay rate are: (i) the orthogonality between $\mathbf{H}$ and $\mathbf{S}$, as enforced by the optimization described in Equation (3), and (ii) the estimation error  $\mathrm{N}(\mathbf{H}\mathbf{x}^{\*})$ of the network $\mathrm{G}^*(\mathbf{y})$.
>
> **Theorem (PnP-NPN Convergence)**
> Consider the fidelity term $g(\tilde{\mathbf{x}}) = \Vert\mathbf{y} - \mathbf{H}\tilde{\mathbf{x}}\Vert_2^2$, and assume the denoiser $D_\sigma$ satisfies Assumption 2. Let the matrix $\mathbf{S}$ be constructed according to Equation (3) and satisfy the RIP condition (Definition 2) with constant $\Delta_{{M}}^S \in [0,1)$. Suppose the network $\mathrm{G}^*$ satisfies Assumption 3. Then, for a finite number of iterations $\ell = 1, \dots, L_\phi$ within the CIZ, the residual $\Vert\mathbf{x}^{\ell+1} - \mathbf{x}^\ast\Vert_2^2$ decays linearly at a rate:
> $$
> \rho \triangleq (1+\delta) \left( \left\Vert \mathbf{I} - \alpha (\mathbf{H}^{\top} \mathbf{H} + \mathbf{S}^{\top} \mathbf{S}) \right\Vert_2^2 + (1+\Delta_{\mathcal{M}_D}^S) \Vert \mathbf{S} \Vert_2^2 \right).$$
>
> ---
> ## W2. Assumptions in theoretical derivations
> Thanks for these interesting questions. The step size $\alpha$ was chosen to $\alpha = 1/\Vert\nabla^2 g(\mathbf{x})\Vert_2^2$ following theory on gradient descent [1]. Additionally, in the compressed sensing theory, a wide range of studies have been performed on computing RIC for random Bernoulli or Gaussian matrices (the ones used in CS experiments) [2], proving that RIC is around 0.1-0.3 with high probability.
>
> [1] Notes for Optimization Algorithms Spring 2023 (2023).
>
> [2] Compressive sensing and structured random matrices (2010).
>
> ---
> ## W3. Convergence Improvement Zone (CIZ)
> Empirical evidence provided in Fig. 2 confirms that the CIZ is non-empty. This zone is important because, within it, the learned regularizer contributes meaningful reconstruction information, enhancing convergence and accuracy. For clarity, we present numerical values extracted from Fig. 2(c) in the Table below. The network estimation error was $\Vert\mathrm{N}(\mathbf{H}\mathbf{x}^\ast)\Vert_2^2=  0.0584$. The CIZ, therefore, corresponds to iterations where \$\Vert\mathbf{Sx}^\ell - \mathbf{Sx}^\ast\Vert_2^2 > 0.0584\$. Representative iterations from the algorithm were selected, evaluating projection error $||\mathbf{Sx}^\ast - \mathbf{Sx}^\ell ||_2^2$ for NPN and baseline methods. As shown, when projection error exceeds network estimation error, NPN demonstrates consistently lower convergence ratios and higher PSNR values compared to baseline, indicating superior convergence performance.
>
> |\$\ell\$ | NPN $\Vert\mathbf{Sx}^\* - \mathbf{Sx}^\ell\Vert_2^2\$ | Base $\Vert\mathbf{Sx}^\* - \mathbf{Sx}^\ell\Vert_2^2\$ | NPN PSNR | Base PSNR  | Base Convergence Ratio|NPN Convergence Ratio|
> | -------- | ------- | -----|---|---|---|------|
> |5|5.124|5.35|5.96|5.82|0.9970|0.9938|
> |50|0.814|2.80|12.53|8.13|0.9914|0.9717|
> |75|0.077|1.48|18.65|10.32|0.9885|0.9822|
> |150|0.040|0.09|20.82|18.08|0.9927|0.9995|
>
> The following table presents results for the deblurring task with a Gaussian kernel ($\sigma = 2.0$) using PnP-ADMM and a DnCNN denoiser. In this case, since the estimation error of the network is quite low, $\Vert\mathrm{N}(\mathbf{H}\mathbf{x}^\ast)\Vert_2^2 = 1.37×10⁻⁴$,  the CIZ covers all the algorithm iterations. Thus, the NPN method consistently achieves better convergence ratios and yields improved reconstructions in all iterations.
>
>  $\ell$ | NPN $\Vert\mathbf{Sx}^\ast - \mathbf{Sx}^\ell\Vert_2^2$| Base $\Vert\mathbf{Sx}^\* - \mathbf{Sx}^\ell\Vert_2^2$| NPN PSNR | Base PSNR  | Base Convergence Ratio| NPN Convergence Ratio |
> |-----|----|-----|----|-----|---|----|
> | 5|5.51×10⁻⁴|8.44×10⁻⁴| 30.84|29.56|0.9932|0.9774|
> | 25|3.81×10⁻⁴|8.70×10⁻⁴| 32.00|29.54|1.0007|0.9983|
> | 50|3.57×10⁻⁴|8.79×10⁻⁴| 32.21| 29.47|1.0000|0.9995|
> | 75|3.49×10⁻⁴|8.66×10⁻⁴| 32.32| 29.49|0.9999|0.9995|
>
> Based on these results, we conclude that the CIZ is not empty and is achievable in practice. Additionally, analysis demonstrates that an effective estimation of the network $\mathrm{G}$ provides larger improvement margins.
>
> ---
> ## W4. Additional reconstruction methods
> We appreciate your suggestion. We consider expanding the application of the proposed method to two new reconstruction frameworks.
> 1. Deep image prior (DIP) [1] reconstruction framework for the deblurring task using a new dataset, Places365.
> 2. Two diffusion models: reconstruction frameworks, DPS [2] and DiffPIR [3], for limited-angle CT.
>
> ### Deep Image Prior
> We study the deblurring task using the Places365 dataset, with 28,000 training and 7,000 testing images, all resized to 128×128. The DIP framework uses untrained neural networks to reconstruct signals from measurements $\mathbf{y}$. However, DIP may overfit measurement noise. Incorporating NPN regularization mitigates this by promoting consistency. The resulting optimization problem is:
> $$\mathrm{K}^\ast = \arg\min_{\mathrm{K}} \Vert\mathbf{y}-\mathbf{H}\mathrm{K}(\mathbf{z})\Vert_2^2 + \gamma \Vert\mathrm{G}^\ast(\mathbf{y})-\mathbf{S}\mathrm{K}(\mathbf{z})\Vert_2^2, \quad \hat{\mathbf{x}} = \mathrm{K}^\ast(\mathbf{z}),  \quad \mathbf{z}\sim\mathcal{N}(0,1).$$
> The following table shows the performance for different values of $\gamma$, showing improvements of up to 5 dB in reconstruction quality for 10 images of the test set of the Places365 dataset.
> |$\gamma$|0.0(Base)|0.001|0.01|0.1|0.5|1.0|
> |----|----|--|----|---|---|---|
> |PSNR|17.59±5.2|18.79±2.9|_21.93±1.5_| **22.16±1.1**| 21.19±1.9|20.54±2.1|
> |SSIM|0.31±0.07| 0.28±0.11| 0.49±0.13| **0.54±0.10**|_0.5±0.14_|0.48±0.17|
>
> ---
> ### Diffusion Models (DMs)
>
> We consider two types of DMs: DPS [2] and DiffPIR [3]. These frameworks have become standard in solving imaging inverse problems with diffusion priors. In DPS, a measurement consistency function is used to condition the generative process. We incorporate the NPN regularization into the posterior sampling step. For every time-step $t$, we modify the posterior sampling step as:
> $$ \mathbf{x}^{t-1} ={\mathbf{x}^{t-1}} - \zeta_\ell \nabla_{\mathbf{x}^t} \left(\left\Vert \mathbf{y} - \mathbf{H}\hat{\mathbf{x}}_0 \right\Vert_2^2 + \gamma \Vert\mathrm{G}(\mathbf{y}) -\mathbf{S}\hat{\mathbf{x}}_0\Vert_2^2 \right)$$
> In DiffPIR, we modify the data proximal subproblem with:
> $$\hat{\mathbf{x}}_0^{(t)} =\underset{\mathbf{x}}{\mathrm{argmin}} \left\Vert \mathbf{H}\mathbf{x} - \mathbf{y} \right\Vert_2^2 + \rho_t \left\Vert \mathbf{x} - \mathbf{x}_0^{(t)} \right\Vert_2^2 + \gamma \left\Vert \mathbf{S}\mathbf{x} - \mathrm{G}(\mathbf{y}) \right\Vert_2^2$$
>
> We train the DM for 1000 epochs, batch size 4, using AdamW with learning rate $3×10⁻⁴$. Noise schedule: cosine from $\beta_1=10⁻⁴$ to $\beta_T=0.02,  T=1000 $. Training data: 10,368 slices from the LoDoPaB-CT dataset, resized to 256x256. Evaluation on 10 test slices.
> | $ \gamma$| NPN–DPS | NPN–DiffPIR |
> |-----|-----|----|
> |0.0(Baseline)|28.22|_31.30_|
> |10⁻⁴|28.30|**31.91**|
> |10⁻³|28.47|30.53|
> |0.1|_30.06_|28.98|
> |0.2|**30.07**|28.57|
> |0.5|29.90|28.0|
>
> [1] Deep image prior. CVPR 2018.
>
> [2] Diffusion Posterior Sampling for General Noisy Inverse Problems. ICLR 2023.
>
> [3] Denoising diffusion models for plug-and-play image restoration. CVPR 2023.
>
> ---
> ## W5. Additional PnP denoisers
>
> We thank the reviewer for the comments. Section A.8 of the supplementary material presents results for various $\gamma$ values across SPC, MRI, and deblurring tasks, showing consistent improvements over the baseline and demonstrating robustness to the regularization parameter. We also replaced FISTA with ADMM in the PnP framework, separating data fidelity and prior subproblems, and evaluated several denoisers: Restormer [1], DnCNN [2], DnCNN-Lipschitz [3], and DRUNet [4]. For deblurring with a Gaussian kernel ($\sigma = 2.0$), we ran 200 ADMM iterations with step size $\alpha = 0.5$ and $\gamma = 0.7$. Results show that our regularization improves performance across all tested denoisers.
> | Method| Restormer | DnCNN | DnCNN-Lipschitz | DRUNet | Sparsity|
> |---|---|----|-----|---|--|
> | Baseline| 29.86| 29.55 | 30.36| 29.68  | 28.75|
> | NPN| **32.62** | 32.12 | _32.35_| 32.07| 29.75|
>
> Hyperparameters λ₁ = 0.001 and λ₂ = 0.01, which were found to be optimal, did not significantly affect performance. The most influential hyperparameter is $\gamma$, the regularizer weight, as demonstrated in DMs, PnP, and DIP reconstruction frameworks.
>
> [1] Restormer: Efficient Transformer for High-Resolution Image Restoration. CVPR 2022.
>
> [2] Beyond a Gaussian denoiser: Residual learning of deep CNN for image denoising. IEEE TIP 2017.
>
> [3] Plug-and-Play Methods Provably Converge with Properly Trained Denoisers. ICML 2019.
>
> [4] Plug-and-Play Image Restoration with Deep Denoiser Prior. IEEE TPAMI 2021.

---

> > ### Comment · Reviewer_PAt7 · 2025-08-01
> > **Thanks for the answers**
> >
> > ## W2
> >
> > Thanks for the clarifications. It would also be valuable to have the authors opinion on whether operators different from compressed sensing (deblurring, inpainting, MRI, ...) meet Theorem 1 assumptions.
> >
> > ## W3
> >
> > My initial concern was that Figure 2 gave insights on the existence of the CIZ for one problem in particular. It would be interesting to know to what extent this area exists for a broader set of problems, given that its existence is key to the the theoretical results.
> >
> > ## Other points
> >
> > I thank the authors for the new experiments and explanations, which add value to the paper.
> >
> >
> > My remaining concern is that it's still not clear to which inverse problems the theoretical results can be applied (W2 and W3), and it's an important point to give more strength to the theory. If the assumptions are met only for compressed sensing, it would be great if the authors could make it clearer in the paper. Otherwise, I think the paper needs extra experiments to illustrate that Theorem 1 and 2 assumptions are met on a regular basis in other problems.

---

> > > ### Author Response · Authors · 2025-08-04
> > > **Response to reviewer**
> > >
> > > Thanks for your follow-up questions.
> > >
> > > ## W2
> > >
> > > We want to clarify that the most important assumption in Th. 1 is the RIP. Several inverse problems (IPs) satisfy the RIP-type assumptions, or admit close analogs that justify their application. In particular:
> > >
> > > a) In MRI (undersampled Fourier matrices), the RIP holds with high probability when the number of measurements is of the order of $\log\log n$ as [1].
> > >
> > > b) In image deblurring (ID), similar analyses have been done for compressed sensing (CS)-like recovery guarantees [2], ensuring that the convolution kernel should retain the relative intensity of the clean image (which is achieved with Gaussian kernels).
> > >
> > > c) On the other hand, image inpainting can be analyzed from a matrix completion (MC) point of view, where several works have been done on solving MC using sparse recovery, requiring that with high probability the underlying sampling operator satisfies the RIP with a bounded constant. [3]
> > >
> > > RIP-like analysis has also been performed for other imaging modalities such as spectral imaging [4] and phase retrieval [5], among others. These examples demonstrate that the key assumption in Th. 1 is not limited to classical CS, supporting the broader applicability of the CIZ observed empirically.
> > >
> > > [1] Sparse reconstruction by convex relaxation: Fourier and Gaussian measurements. 2006
> > >
> > > [2] Compressive deconvolution in random mask imaging. 2015
> > >
> > > [3] CS and MC with constant proportion of corruptions. 2013
> > >
> > > [4] Restricted isometry property in coded aperture compressive spectral imaging. 2012
> > >
> > > [5] A strong restricted isometry property, with an application to phaseless CS. 2016
> > >
> > > ---
> > >
> > > ## W3
> > >
> > > We thank the reviewer for this question. The CIZ was originally illustrated in Fig. 2 for CS; here, we extend the empirical evidence to additional IPs to clarify its prevalence. In particular, we evaluate the behavior in CT and MRI using PnP-FISTA and the NPN regularization.
> > >
> > > For CT (Tab. 1), the residual projection error (PE) drops below the network estimation error (NEE) ($\Vert \mathrm{N}(\mathbf{Hx}^{\*})\Vert = 6.12$) around iteration 90. Although the convergence ratio does not improve substantially in this setting, the NPN-regularized method still yields a notable recovery quality gain over the baseline. In MRI (Tab. 2), the PE becomes smaller than the NEE ($\Vert \mathrm{N}(\mathbf{Hx}^{\*})\Vert = 1.211×10⁻⁴$) near iteration 850, and we observe sustained convergence improvement up to iteration 1016, together with a significant boost in recovery accuracy compared to baseline PnP. These results indicate that the CIZ phenomenon appears in multiple modalities.
> > >
> > > **Table 1.**  CT experiments.
> > >
> > > |$\ell$|NPN $\|\mathbf{Sx}^{\*} - \mathbf{Sx}^\ell\|$|Base $\|\mathbf{Sx}^{\*} - \mathbf{Sx}^\ell\|$|NPN PSNR (dB)|Base PSNR (dB)|NPN Convergence Ratio|Base Convergence Ratio|
> > > |--:|--:|--:|--:|--:|--:|--:|
> > > |10|3118.010|429986.312|0.578|-15.279|1.060|0.655|
> > > |50|6.947|105.221|22.631|13.718|0.929|0.972|
> > > |90|5.161|79.845|25.002|16.891|0.995|0.995|
> > > |130|5.119|75.937|25.312|17.286|0.999|0.999|
> > >
> > > ---
> > >
> > > **Table 2.** MRI experiments.
> > >
> > > |$\ell$|NPN $\|\mathbf{Sx}^{\*} - \mathbf{Sx}^\ell\|$|Base $\|\mathbf{Sx}^{\*} - \mathbf{Sx}^\ell\|$|NPN PSNR (dB)|Base PSNR (dB)|NPN Convergence Ratio|Base Convergence Ratio|
> > > |--:|--:|--:|--:|--:|--:|--:|
> > > |50|$2.6779×10⁻⁴$|$2.6888×10⁻⁴$|35.72|35.70|0.9998|0.9999|
> > > |400|$2.0365×10⁻⁴$|$2.5033×10⁻⁴$|36.91|36.01|0.9988|0.9997|
> > > |800|$1.2545×10⁻⁴$|$2.1661×10⁻⁴$|39.01|36.64|0.9992|0.9997|
> > > |1200|$1.1157×10⁻⁴$|$1.9903×10⁻⁴$|39.52|37.01|1.0000|0.9999|
> > >
> > > These additional modalities, together with those in the main manuscript and earlier parts of the rebuttal, support that the CIZ is not an isolated artifact but recurs across different IPs, with variability in onset and strength.
> > >
> > > ---
> > >
> > > ## Other points
> > >
> > > We thank the reviewer for these important points, which have helped sharpen both the theoretical framing and empirical validation. Indeed, the proposed method has great potential: our analysis is built on RIP-type conditions, and the cited literature shows that those conditions hold beyond classical CS—for example, in MRI, ID under appropriate kernel conditions, spectral imaging, and phase retrieval. In the manuscript, we showed the existence and behavior of the CIZ empirically for CS (Fig. 2). Previously, in the rebuttal (see W3), we showed the results for ID. Here (see W3), we show additional results for MRI and CT. These results support that the assumptions of Ths. 1 and 2 are satisfied in practice in multiple settings and clarify the regimes where the improvements manifest. In the final version, we will (a) explicitly enumerate which IP classes satisfy the key assumptions, citing the relevant prior work for each, (b) clearly separate the theoretical guarantee from the empirical observation, and (c) include the new experiments and discussion so that the scope, applicability, and limitations of the approach are stated with appropriate precision.

---

> > > > ### Comment · Reviewer_PAt7 · 2025-08-04
> > > > **Thanks for the clarification**
> > > >
> > > > I thank the authors for their detailed answers to my questions, my concerns have been adressed and I recommend including all these details in the final version of the paper.

---

> > > > > ### Author Response · Authors · 2025-08-05
> > > > > **Concluding remarks to reviewer**
> > > > >
> > > > > Thank you for your detailed feedback and your kind words about our use of null‐space structure and the clarity of our presentation. Your suggestions have guided us in adding new experiments, deeper analyses, and the requested clarifications, all of which will appear in the final manuscript. We sincerely appreciate the time you invested and the constructive discussion, which has significantly strengthened our work.

---

### Official Review · Reviewer_hHJq · 2025-06-27

**Clarity:** 2
**Significance:** 3
**Originality:** 3
**Rating:** 5
**Confidence:** 3

**Summary:**

The paper proposes a new class of regularization, referred to as non-linear projections of the null space (NPN). Assuming a linear inverse problem with a sensing matrix $H$, the basic idea of NPN is to find a proper range-null space decomposition. In particular, NPN jointly searches the projection matrix and neural network restricted to the null space of $H$. The authors theoretically analyze the convergence property of the plug-and-play (PnP) algorithm with NPN and numerically demonstrate that it can converge faster than a naive PnP algorithm. Additionally, they demonstrate some image reconstruction experiments, suggesting the superiority of NPN.

**Questions:**

* The reviewer wonders whether the convergence of PnP-NPN is possible outside the convergence improvement zone. Although the reviewer understands that the proof of Thm. 1 requires the zone, it is preferable to analyze the convergence property of the algorithm without it, as noted above.

* Although NPN successfully improves performance, it requires additional training parameters and a training process. The authors will need to compare it with existing methods in terms of computational time and/or the number of parameters.

**Ethical Concerns:**

["NO or VERY MINOR ethics concerns only"]

**Final Justification:**

The theoretical part of the manuscript has been significantly improved thanks to the reviewers’ comments. Although the proposed method enhances convergence performance only over a finite number of iterations, the numerical results demonstrate substantial performance gains. Therefore, the reviewer considers the manuscript’s contribution to be sufficiently valuable for acceptance at the conference.

**Limitations:**

Yes. However, the authors might need to add some discussions on limitations based on the additional experiments or analyses suggested by reviewers.

**Paper Formatting Concerns:**

I have no concerns.

**Quality:**

3

**Strengths And Weaknesses:**

**Strength**

* The paper proposes a new regularization technique. Although the range-null decomposition has been studied in the literature, the idea of NPN seems novel. (Quality)

* The authors present both theoretical analysis and numerical demonstrations, which effectively demonstrate the superiority of NPN. (Significance)

* As described in the paper, NPN is not limited to the PnP algorithm. The idea presented in this paper will appeal to a wide range of readers in machine learning and signal processing. (Significance)

**Weakness**

* The statement of Thm. 1 is confusing. Thm. 1 claims that the algorithm converges to a fixed point within the (finite) convergence improvement zone. However, the authors only show in the proof that the residual error decreases in the zone, which seems inadequate for showing convergence to a fixed point. A similar expression appears in l. 263: "Theorem 1 guarantees convergence of the sequence $\{\mathrm x^l\}$ to a fixed point,"  but convergence will occur only if the zone is infinite. (Clarity, Quality)

* In Sec. 5.1, the authors define the convergence improvement zone of two figures (Fig.2b and 2c) in different ways. Although both zones look to coincide, are the definitions accurate? (Clarity)

* There are several typos and inaccuracies in the notations. Please check the manuscript carefully. (Clarity)

e.g.)

> l.159: (RNSD) -> RNSD

> l.160: $\tilde x=\dots$ seems intuitively understandable but inaccurate.

> l.172: $\mathcal F=f_1^T\dots$ needs $``\dots''$.

> l.194: $A=[H^T,S^T]$ should be $A=[H^T,{\tilde S}^T]$.

> l.320: "Figure 2(b)" should be "2b)."

> l.876 in Appendix: "according to ??" should be revised.

> Alg. 2-5 in Appendix: $\frac{\sqrt{1+4t')^2}}{2}$ should be $\frac{\sqrt{1+4(t')^2}}{2}$

* Although PnP-NPN algorithms are given in Appendix as pseudocode, the details of the network architecture of $\mathrm G$ are missing. It lacks the reproducibility of the paper. (Clarity)

---

> ### Author Rebuttal · Authors · 2025-07-31
>
> We appreciate the reviewer’s constructive feedback, which has led to a quality improvement of the paper. We also thank the reviewer for recognizing our work as "appealing to a wide range of readers in machine learning and signal processing."
> Below, we address each concern in detail.
>
> # Weaknesses
>
> ## W1. Clarity on Theorem 1
>
> We thank the reviewer for the insightful comment and agree that the fixed-point convergence claim in the original statement of Theorem 1 may be misleading. Our main contribution in this result is not to prove convergence to a unique fixed point, but rather to establish a linear residual decay rate within a bounded number of iterations, defined as the *convergence improvement zone* (CIZ):
>
>
> $$
> \mathcal{L}_\phi = \{ \ell :  \Vert \mathrm{N}(\mathbf{H}\mathbf{x}^*)\Vert_2^2 < \Vert\mathbf{S}\mathbf{x}^\ell - \mathbf{S}\mathbf{x}^\ast \Vert_2^2 \}
> $$
>
> Specifically, Theorem 1 shows that the error $\Vert\mathbf{x}^{\ell+1} - \mathbf{x}^\ast\Vert_2^2$ decreases at a linear rate:
>
> $$
> \rho \triangleq (1+\delta) \left( \left\Vert \mathbf{I} - \alpha (\mathbf{H}^{\top} \mathbf{H} + \mathbf{S}^{\top} \mathbf{S}) \right\Vert_2^2 + (1+\Delta_{\mathcal{M}_D}^S) \Vert \mathbf{S}\Vert_2^2\right),
> $$
>
> with respect to the previous iteration error $\Vert\mathbf{x}^{\ell} - \mathbf{x}^\ast\Vert_2^2$. Here, the dominant terms controlling the decay rate are: (i) the orthogonality between $\mathbf{H}$ and $\mathbf{S}$, as enforced by the optimization described in Equation (3), and (ii) the estimation error  $\mathrm{N}(\mathbf{H}\mathbf{x}^{\*})$ of the network $\mathrm{G}^*(\mathbf{y})$. This rate demonstrates improved stability relative to standard PnP schemes, particularly when the term $\Vert\mathbf{I} - \alpha(\mathbf{H}^{\top}\mathbf{H})\Vert_2^2$ is large.
>
> To avoid confusion, we have revised the theorem statement to explicitly remove any misleading fixed-point claim, clearly specifying that the result holds for a finite number of iterations $ L_\phi$:
>
> **Theorem (PnP-NPN Convergence)**
> Consider the fidelity term $g(\tilde{\mathbf{x}}) = \Vert\mathbf{y} - \mathbf{H}\tilde{\mathbf{x}}\Vert_2^2$, and assume the denoiser $D_\sigma$ satisfies Assumption 2. Let the matrix $\mathbf{S}$ be constructed according to Equation (3) and satisfy the RIP condition (Definition 2) with constant $\Delta_{{M}}^S \in [0,1)$. Suppose the network $\mathrm{G}^*$ satisfies Assumption 3. Then, for a finite number of iterations $\ell = 1, \dots, L_\phi$ within the CIZ, the residual $\Vert\mathbf{x}^{\ell+1} - \mathbf{x}^\ast\Vert_2^2$ decays linearly at a rate:
>
> $$
> \rho \triangleq (1+\delta) \left( \left\Vert \mathbf{I} - \alpha (\mathbf{H}^{\top} \mathbf{H} + \mathbf{S}^{\top} \mathbf{S}) \right\Vert_2^2 + (1+\Delta_{\mathcal{M}_D}^S) \Vert \mathbf{S} \Vert_2^2 \right).
> $$
>
> ---
> ## W2. Convergence improvement zones
>
> We thank the reviewer for pointing this out. The CIZ is defined as the region in which the network estimation error satisfies $\mathcal{L}_\phi = \{\ell :\Vert\mathrm{N}(\mathbf{H}\mathbf{x}^{\ast})\Vert_2^2 <\Vert\mathbf{Sx}^\ell - \mathbf{Sx}^{\ast}\Vert_2^2\}$. The drawn zones in Fig. 2b) and Fig. 2c) should match as the algorithm, as Fig. 2c) is the theoretical definition of the CIZ, and Fig. 2b) is the empirical validation of the improvement in the convergence ratio.
>
> In Fig. 2b, we illustrate the behavior of the algorithm using the empirical convergence ratio $\Vert\mathbf{x}^{\ell+1} - \mathbf{x}^\ast\Vert_2^2 / \Vert\mathbf{x}^\ell - \mathbf{x}^\ast\Vert_2^2$. We acknowledge an error in the graphical annotation of the convergence zone: for instance, in the blue curve, the proposed regularized algorithm with $\mathbf{S}$ designed in Eq. (3), the zone should terminate around iteration $\ell = 60$, where the black curve (baseline algorithm) drops below the blue curve. Similarly, for the red curve, $\mathbf{S}$  (QR), the zone should go until around $\ell = 120$.
>
> In Fig. 2c, the CIZ is defined based on the inequality $\mathcal{L}_\phi = \{\ell :\Vert\mathrm{N}(\mathbf{H}\mathbf{x}^\ast)\Vert_2^2 <\Vert\mathbf{Sx}^\ell - \mathbf{Sx}^{\ast}\Vert_2^2\}$, indicated by the intersection of the solid and dotted lines. The key observation across both figures is that the convergence of the algorithm is enhanced in the regime where the estimation error of the network is smaller than the projection error, indicating that the learned regularization is providing useful information to guide the recovery process. With the correction mentioned earlier, the zones in both figures align more closely, validating the developed theory.
>
> ---
> ## W3. Typos in the manuscript
>
> We appreciate the reviewer's careful reading of the paper. We will fix the raised typos in the final version of the paper.
>
> ---
> ## W4. Network architecture details
>
> We appreciate the reviewer’s concern regarding clarity and reproducibility. We would like to kindly clarify that Section A.5 of the Supplementary Material contains the architecture details of the network $\mathrm{G}$ used in all experiments. To improve clarity, we will explicitly state in the revised Supplementary that this section refers to $\mathrm{G}$. However, these are the model specifications used in the experiments.
>
>  **SPC**: For the single‐pixel camera experiments, we employ a ConvNeXt‐inspired backbone [1]. The network has five ConvNeXt blocks, each comprising two successive $3\times3$ convolutions with ReLU activations. A final $3\times3$ convolutional layer projects the 128 features back to one channel. The output feature map is flattened and passed through a linear module to match the measurement dimensionality. We use cosine‐based positional encoding with a dimension of 256, and set the number of blocks to 5 and the base channel width to 128.
>
> **MRI, Deblurring and CT**: To train $\mathrm{G}$ for the MRI and Deblurring experiments, we use a U-Net architecture [2] with three downscaling and three upscaling modules. Each module consists of two consecutive Conv → ReLU blocks. The downscaling path uses max pooling for spatial reduction, starting with 128 filters and increasing up to 1,024 at the bottleneck. The upscaling path performs nearest-neighbor interpolation before each module, progressively reducing the number of filters. A final 2D convolutional layer without activation produces the output. Skip connections link corresponding layers in the encoder and decoder.
>
> [1] Zhuang Liu, et al. A ConvNet for the 2020s. In CVPR 2022.
>
> [2] Olaf Ronneberger, et al. U-net: Convolutional networks for biomedical image segmentation. In MICCAI 2015.
>
> ---
> # Questions
>
> ## Q1. Convergence without an improvement zone
>
> We thank the reviewer for raising this important point. The convergence of PnP-NPN outside the CIZ can indeed be studied under the general theoretical frameworks developed in the PnP literature. For instance, [1, Theorem 1] for PnP-projected gradient descent and [2, Theorem 1] for PnP-ADMM provide convergence guarantees under assumptions that are also satisfied in our setting, such as boundedness of the denoiser and Lipschitz continuity of the data fidelity gradient.
>
> However, our inclusion of the CIZ serves to highlight a specific regime in which the proposed NPN regularization actively contributes meaningful signal information — particularly in the form of structured null-space projections. Within this zone, we can rigorously characterize an improved convergence rate due to the alignment between the learned prior and the null-space component of the signal.
>
> While convergence outside this zone is still covered by general PnP theory, our main contribution lies in identifying and exploiting this beneficial regime where convergence is not only guaranteed but accelerated due to the informative nature of the learned null-space prior.
>
> [1] Provable preconditioned plug-and-play approach for compressed sensing MRI reconstruction. IEEE TCI, 2024.
>
> [2] Plug-and-play ADMM for image restoration: Fixed-point convergence and applications. IEEE TCI, 2016.
>
> ---
>
> ## Q2. Comparison in terms of computational time and number of parameters
>
> This is a very nice question. To create a fair comparison with the proposed method, we consider the network $ W^* : \mathbb{R}^m \to \mathbb{R}^n$  which has the same number of parameters as $G$ but is trained to reconstruct directly the underlying signal, i.e. $ W^* = \arg\min_{W}\ \mathbb{E}_{\mathbf{x},\mathbf{y}}\big\Vert\mathbf{x} - W(\mathbf{y})\big\Vert_2^2.$
>
> The network was trained for 300 epochs (same as $G$) using the AdamW optimizer with learning rate $1\times10^{-4}$. We evaluate two aspects:
>
> 1. The reconstruction performance of the network $W$.
> 2. Incorporation into PnP-ADMM of the regularization in the form  $$
>    \phi_{W}(\mathbf{x}) = \big\Vert\mathbf{S}\mathbf{x} - \mathbf{S}W^*(\mathbf{y})\big\Vert_2^2.
>    $$
>
> We trained $W$ on the Places365 dataset. PnP-ADMM was run for 100 iterations with the DnCNN-Lipschitz denoiser, setting $\alpha = 0.3$ and $\gamma = 0.2$.  The results show that the proposed approach overall improves the baseline,  the reconstruction of the model $\mathrm{W}$, and the regularization based on this model.
>
> | Metric| Base PnP-ADMM | NPN PnP-ADMM | PnP-ADMM with $\phi_{W}$ | $W^*(\mathbf{y})$ |
> |--------------:|--------------:|-------------:|-----------------------:|------------------:|
> |PSNR|_22.30_ |**23.87** |21.20 |19.71 |
> |SSIM|_0.586_|**0.678** |0.534|0.490|

---

> ### Comment · Reviewer_hHJq · 2025-08-02
>
> First, I appreciate the authors' response.
>
> **W1, W2 & Q1**
>
> Now, I figure out the meaning of CIZ and the convergence property outside of CIZ. The convergence property of the proposed algorithm irrelevant to CIZ is crucial. So, I recommend that the authors also mention it as well as convergence improvement in the manuscript.
>
> Another question: according to [1], the convergence rate outside of CIZ is also linear. The difference lies in the convergence rate. However, it is not intuitive that the "improved rate" $\rho$ is smaller than the standard rate in [1] because of the term $(1+\Delta_{\mathcal M_D}^S)\|S\|$. Is the improved rate always smaller than the standard rate (although it is numerically verified in the experiments)?
>
> [1] Provable preconditioned plug-and-play approach for compressed sensing MRI reconstruction. IEEE TCI, 2024.
>
> **Q2**
>
> Thank you for showing another result. How about training and execution time in this case?

---

> > ### Author Response · Authors · 2025-08-04
> > **Response to reviewer**
> >
> > Thanks for your follow-up questions.
> >
> > >**W1, W2 & Q1**
> >
> > We thank the reviewer for this suggestion. We will explicitly add to the manuscript a paragraph after Theorem 1 that clarifies two points: (i) the algorithm still converges linearly outside the CIZ, and (ii) the CIZ defines a localized region of accelerated convergence—i.e., a quantified “convergence improvement” over the baseline rate. The added text will distinguish the global linear guarantee from the faster residual decay inside the CIZ and introduce the improvement metric used to compare the two regimes.
> >
> > >Is the improved rate always smaller than the standard rate (although it is numerically verified in the experiments)?
> >
> > We appreciate the request for clarification. The gain on the convergence rate $\rho$ of the proposed method is considerable when there is a small estimation error on the network, which is the term upper-bounded by $(1-\Delta_{M_D}^S)\Vert\mathbf{S}\Vert$, since $\Vert\mathbf{I}-\alpha(\mathbf{H^\top H + S^\top S})\Vert\ll \Vert\mathbf{I}-\alpha(\mathbf{H^\top H })\Vert$ (bound obtained with traditional PnP convergence analysis [1]) due to the orthogonality between $\mathbf{H}$ and $\mathbf{S}$, which is achieved by the design of $\mathbf{S}$ (section 3.1).
> > In this work, we provided numerical examples to validate the assumption on the bounds of the network estimation error; however, future work will focus on guarantees on the estimation error based on the structure $\mathbf{H,S}$ and neural network architecture $\mathrm{G}$.
> >
> > [1] Provable preconditioned plug-and-play approach for compressed sensing MRI reconstruction. IEEE TCI, 2024.
> >
> > >**Q2**: How about training and execution time in this case?
> >
> > Thanks for this question. In the tables below, we show the training and execution time in seconds (s) and minutes (min), respectively. These results are obtained for a batch size of 100 images with a resolution of $128 \times 128$ and 100 iterations of the PnP-ADMM. $\mathrm{W}$ (reconstruction network from the measurements $\mathbf{y}$) and $\mathrm{G}$ are trained for 300 epochs. Recall that $\phi_{\mathrm{W}}(\mathbf{x}) = \Vert \mathbf{S}\mathrm{W}(\mathbf{y}) - \mathbf{Sx}\Vert$.
> >
> > | Method | Training time (min) |
> > |---|---|
> > | $\mathrm{G}(\mathbf{y})$ | 385 |
> > | $\mathrm{W}(\mathbf{y})$ | 390 |
> > *Table 1: Training time for the networks $\mathrm{W}$ and $\mathrm{G}$ (NPN).*
> >
> > | Method | Execution time (s) |
> > |---|---|
> > | Base PnP-ADMM | **12.48** |
> > | NPN PnP-ADMM | *12.68* |
> > | PnP-ADMM with $\phi_{W}$ | 12.96 |
> > *Table 2: Execution times for PnP with network $\mathrm{W}$ and $\mathrm{G}$ (NPN).*
> >
> > The training times of $\mathrm{W}$ and $\mathrm{G}$ are very similar due to having the same number of parameters. Despite a modest increase of 0.20 seconds over the baseline PnP-ADMM, NPN achieves a 1.57 dB PSNR gain. Moreover, compared to PnP-ADMM with $\phi_{\mathrm{W}}$, NPN is 0.28 seconds faster while delivering a 2.67 dB improvement in PSNR (see Table from Q2). These results show that the NPN regularization yields substantial quality gains with minimal or favorable time trade-offs.

---

> > > ### Comment · Reviewer_hHJq · 2025-08-04
> > >
> > > Thank you for your additional response. I believe all of my concerns have now been addressed.
> > > I hope my comments were helpful in improving the quality of your manuscript.

---

> > > > ### Author Response · Authors · 2025-08-05
> > > > **Concluding remarks to reviewer**
> > > >
> > > > Thank you for your constructive feedback and kind words on NPN’s novelty, theoretical analysis, and broad applicability. We are delighted to have addressed all of your concerns. Your comments have directly led us to strengthen the manuscript by adding new results, deeper analyses, and the requested clarifications, which will all appear in the final version. We sincerely appreciate the time you invested and the thoughtful debate, which has greatly improved the quality of our work.

---

### Official Review · Reviewer_RKuN · 2025-06-29

**Clarity:** 3
**Significance:** 3
**Originality:** 3
**Rating:** 4
**Confidence:** 3

**Summary:**

The method proposes to learn a network to estimate the solution’s projection in the null space of the linear operator when solving a linear inverse problem.  It is used as an additional regularization in the optimization (along with the typical prior term) to inform the optimization process about the structure of the linear operator.

**Questions:**

I listed my questions above.  To reiterate, my main questions are about the construction of the low-rank S and the effectiveness of the optimization (3) to recover important subspace that is useful to describe the data distribution. I also have a question about whether in some applications the estimated G(y) would give a trivial solution.

**Ethical Concerns:**

["NO or VERY MINOR ethics concerns only"]

**Final Justification:**

My concerns were (1) how S was chosen, (2) how the network G is able to estimate Sx, (3) insufficient comparison to existing methods.  The rebuttal addresses my concerns.  The authors should add the discussions in the revision.

**Limitations:**

yes

**Paper Formatting Concerns:**

No formatting issue is found.

**Quality:**

3

**Strengths And Weaknesses:**

Strengths
- The paper explains the motivations, assumptions, and the algorithm clearly.

- The paper provides analyses on the convergence of utilizing the proposed regularizer in solving linear inverse problems.


Weaknesses
- Line 141 says the rank of S is p < (n -m), but from Sec 3.1 it seems the full basis of the null space is used.  Is S chosen to be low rank, and if so, the paper does not describe how to choose the low rank S.  In the case of DFT and random matrices, each basis vector seems equal without looking at the data distribution to be solved for.


- Perhaps related to the question above, how do the choice of p and the initialization of S affect the optimization (3)?  When S is initialized as a subset of the null space basis, is the optimization (3) able to get S that is composed of a different set of basis vectors? Also how does the data distribution affect the solution of S?

- I do not fully understand how the network G(y) is able to estimate Sx effectively.  My understanding is y is the measurement, which contains no information about the projection of x in the null space.  In this case, what information does G(y) use to estimate Sx?  For example, in the case of super-resolution, there will be multiple x’s that have the same measurement y but different Sx.  In this case, would G(y) simply estimate E[Sx] of the entire training set, and in the case of super-resolution if we have a large dataset will this expectation simply be 0 and we recover a typical l2 regularization in (2)?


- The comparison in Table 1 is mainly with other null-space methods.  How does the proposed method compare to other types of methods for solving inverse problems, e.g. [1] and [2]?  Can the proposed method be incorporated in these methods?

[1] Solving Linear Inverse Problems Provably via Posterior Sampling with Latent Diffusion Models. NeurIPS 2023.

[2] Solving Inverse Problems with Latent Diffusion Models via Hard Data Consistency. ICLR 2024.

---

> ### Author Rebuttal · Authors · 2025-07-31
>
> We appreciate the reviewer’s constructive feedback, which has helped improve the paper. We also thank the reviewer for noting that the paper provides “analyses on the convergence of utilizing the proposed regularizer in solving linear inverse problems.” Below, we address each concern in detail.
>
> ## Weaknesses
>
> ### W1. Low rank nature of $\mathbf{S}$
>
> We appreciate the reviewer’s comment regarding the rank and construction of the matrix $\mathbf{S}$. The matrix $\mathbf{S} \in \mathbb{R}^{p \times n}$ is not promoted to be low-rank; rather, it is specifically designed to have full row rank. Although we can control the number of rows, choosing $p \leq n-m$, $\mathbf{S}$ does not necessarily span the entire null space basis. Instead, it spans a subspace, which varies depending on the application. For example, in compressive sensing, we promote the maximum rank of $\mathbf{A} = [\mathbf{H}; \mathbf{S}]$, i.e., $m+p \leq n$.
>
> While we do not provide theoretical bounds for the chosen size, we evaluate its effect empirically on reconstruction quality with PnP-FISTA in Figure 8 of the supplementary material. While increasing $p$ may improve performance by capturing more null-space components, it also makes the estimation task of $\mathrm{G}$ more challenging, as it must recover a higher-dimensional projection from measurements $\mathbf{Hx}$. Thus, there is a tradeoff between the rank of $\mathbf{S}$ and the network capacity required for accurate prediction.
>
> In structured settings such as MRI with undersampled DFT matrices, we construct $\mathbf{S}$ using the missing rows of $\mathbf{H}$, which worked well without additional optimization. In contrast, for unstructured sensing matrices (e.g., Bernoulli in compressed sensing), optimizing $\mathbf{S}$ via Equation (3) resulted in better convergence and reconstruction than using a fixed orthonormal basis.
>
> ---
>
> ### W2. Choice of $p$ and data adaptability
>
> The size of $\mathbf{S}$ significantly affects the optimization in Equation (3). A larger $p$ increases the difficulty of projection estimation. Using the same models as in Figure 8, we report projection errors in the table below, showing that increasing the number of rows in $\mathbf{S}$ leads to higher estimation error:
>
> | $p/n$ | $\Vert\mathbf{Sx}^* - \mathrm{G}(\mathbf{y})\Vert_2^2 / \Vert\mathbf{Sx}^*\Vert_2^2 $ |
> |:---:|:------:|
> |0.1|**0.1031**|
> |0.5|_0.1821_|
> |0.9|0.2305|
>
> To illustrate the effect of the data-driven adaptability of $\mathbf{S}$, we clarify the following.  In compressed sensing, to span the null-space, we design a matrix orthogonal to $\mathbf{H}$ via the QR decomposition, and via optimization using the regularization in Eq. (3), which promotes that the rows of $\mathbf{S}$ lie in the null-space of $\mathbf{H}$. To analyze how the data distribution affects the design of $\mathbf{S}$, we optimize four matrices, one for each dataset (MNIST, FashionMNIST, CIFAR10, and CelebA), cross-validating the data-driven invertibility loss $ \mathbb{E}_{\mathbf{x}}\Vert\mathbf{x} - \mathbf{S}^\dagger \mathbf{S}\mathbf{x}\Vert_2^2. $
> The results show that the QR‐initialized matrix provides good invertibility across all datasets, and that optimizing $\mathbf{S}$ via Eq. (3) on each dataset further improves it. Thus, the data distribution can enhance the data‐driven invertibility of an already orthogonal matrix. Testing $\mathbf{S}$ on in‐distribution samples yields the $\textbf{best}$ invertibility ($\textbf{bold}$), while diverse training data (e.g.\ CIFAR10) gives the _second‐best_ (_underlined_), suggesting a single $\mathbf{S}$ could generalize across datasets.
>
> | Test \ S design| QR | MNIST| FashionMNIST| CIFAR10| CelebA|
> |-----|:---:|:----:|:-----:|:-----:|:-----:|
> | MNIST| 9.3×10⁻⁵ | **3.0×10⁻⁵**| _5.0×10⁻⁵_| 5.3×10⁻⁵| 5.6×10⁻⁵|
> | FashionMNIST| 1.8×10⁻⁴ | 7.1×10⁻⁵ | **3.1×10⁻⁵**| _5.1×10⁻⁵_| 5.6×10⁻⁵|
> | CIFAR10| 2.6×10⁻⁴| 1.7×10⁻⁴| 9.0×10⁻⁵| **4.0×10⁻⁵**| _4.5×10⁻⁵_|
> | CelebA| 2.4×10⁻⁴ | 1.6×10⁻⁴| 9.1×10⁻⁵| _4.7×10⁻⁵_| **4.2×10⁻⁵**|
>
> ---
> ### W3. Training $\mathrm{G}$ and implications
>
> We appreciate the reviewer’s thoughtful question. Below, we clarify the role of $\mathrm{G}$ in estimating $\mathbf{Sx}$, particularly given that $\mathbf{y} = \mathbf{Hx}$ contains no explicit information about the null-space component $\mathbf{Sx}$.
>
> 1. Supervised training: The network $\mathrm{G}$ is trained a priori, independently of the reconstruction algorithm, according to Eq (3). The dataset consists of $N$ image samples $ \{\mathbf{x}_i\}$, corresponding measurements $ \{\mathbf{Hx}_i\}$, and null-space projections $\{\mathbf{Sx}_i\}$ for $i = 1, \dots, N$. While $\mathbf{Sx}$ is not directly recoverable from $\mathbf{Hx}$, the dataset enables learning to predict $\mathbf{Sx}$ from $\mathbf{Hx}$ via statistical correlations.
>
> 2. Regularization to avoid trivial solutions: To avoid trivial solutions $\mathrm{G}(\mathbf{y}) \approx \mathbf{0}$ or $\mathbf{Sx = 0}, \mathbf{x}\neq 0$, we include orthogonality and inversion regularizers in the joint training of $\mathbf{S}$ and $\mathrm{G}$ (see Eq. (3)). These regularizers ensure that $\mathbf{S}$ is meaningful and that $\mathrm{G}$ does not collapse.
>
> 3. High-frequency content recovery: In tasks like super-resolution or deblurring, high-frequency details in $\mathbf{x}$ are lost in the measurements $\mathbf{y} = \mathbf{Hx}$ but preserved in $\mathbf{Sx}$. Although this introduces ambiguity, the availability of paired samples $(\mathbf{x}_i, \mathbf{Hx}_i, \mathbf{Sx}_i)$ enables $\mathrm{G}$ to learn plausible mappings. As shown in Figure 3, the trained network $\mathrm{G}^*$ successfully recovers high-frequency components.
>
> ---
> ### W4. Evaluation with diffusion models.
>
> We thank the reviewer for this suggestion. We have integrated our proposed regularization term into two widely adopted diffusion‐model frameworks, Diffusion Posterior Sampling (DPS) [3] and DiffPIR [4]. These frameworks serve as canonical diffusion pipelines upon which newer methods build. Our regularization could likewise be incorporated into other approaches, such as latent‐space diffusion models [1][2]. Specifically, we evaluated the proposed regularization on the limited‐angle CT inverse problem, using 60 out of 180 total views (spaced every 1°) under a parallel‐beam geometry. Here $\mathbf{H}$ corresponds to the undersampled Radon transform matrix. We construct $\mathbf{S}$ using the missing angles of $\mathbf{H}$, i.e., $\mathbf{S}$ contains the remaining 120 views. We train the diffusion model for $1000$ epochs with a batch size of $4$ using the AdamW optimizer and a learning rate of 3×10⁻⁴. We used a cosine variance schedule ranging from $\beta_1=1×10⁻⁴$ to $\beta_T=0.02$ with $T=1000$ time steps.  The training set comprised $10,368$ slices from the LoDoPaB-CT dataset, resized to $256 \times 256$. We evaluated the performance using 10 slices of the test set. The network $\mathrm{G}$ corresponds to a U-Net, and it was trained for 100 epochs with a learning rate of 3×10⁻⁴  with a batch size of $4$ using the AdamW optimizer.
>
> We incorporated the proposed NPN regularization term at line 7 of the DPS algorithm
>
> $$\mathbf{x}^{t-1} = \mathbf{x}^{t-1} - \zeta_\ell \nabla_{\mathbf{x}^t} \left( \left\Vert \mathbf{y} - \mathbf{H}\hat{\mathbf{x}}_0 \right\Vert_2^2 + \gamma \left\Vert\mathrm{G}(\mathbf{y}) - \mathbf{S} \hat{\mathbf{x}}_0 \right\Vert_2^2 \right)$$
> and at line 4 of the DiffPIR algorithm as follows:
> $$\hat{\mathbf{x}}_0^{(t)} = \underset{\mathbf{x}}{\mathrm{argmin}} \left\Vert \mathbf{H}\mathbf{x} - \mathbf{y} \right\Vert_2^2 + \rho_t \left\Vert \mathbf{x} - \mathbf{x}_0^{(t)} \right\Vert_2^2 + \gamma \left\Vert \mathbf{S}\mathbf{x} - \mathrm{G}(\mathbf{y}) \right\Vert_2^2$$
>
> where $\gamma$ is a hyperparameter to weigh the importance of the regularization. The regularization promotes higher consistency in the sampling process.
> We tested both DPS and DiffPIR across several $\gamma$ values. The table below shows the obtained results. For DPS, the NPN regularization consistently improves reconstruction performance by up to 1.85 dB. For DiffPIR, it yields improvements of up to 0.61 dB (for $\gamma = 1×10⁻⁴$). Further exploration of $\gamma$ may yield even greater gains.
>
> | $\gamma$ |NPN–DPS|NPN–DiffPIR|
> |-----|:----:|:--------:|
> | 0.0 (Base)| 28.22| 31.30|
> | 10⁻⁵ | 28.55  |_31.88_|
> |10⁻⁴ | 28.30  | **31.91** |
> | 10⁻³ | 28.47 | 30.53 |
> | 10⁻² | 28.78 | 29.90 |
> | 10⁻¹ |_30.06_| 28.98 |
> | 0.2| **30.07**| 28.57 |
> | 0.5| 29.90| 28.00 |
>
> Additionally, we include the proposed regularization into the deep image prior (DIP) framework [5]. This framework uses untrained neural networks to reconstruct signals from measurements $\mathbf{y}$. However, DIP may overfit measurement noise. Incorporating NPN regularization mitigates this by promoting consistency. The resulting optimization problem is
>
> $$\mathrm{K}^\ast = \arg\min_{\mathrm{K}} \Vert\mathbf{y}-\mathbf{H}\mathrm{K}(\mathbf{z})\Vert_2^2 + \gamma \Vert\mathrm{G}^\ast(\mathbf{y})-\mathbf{S}\mathrm{K}(\mathbf{z})\Vert_2^2, \quad \hat{\mathbf{x}} = \mathrm{K}^\ast(\mathbf{z}),  \quad \mathbf{z}\sim\mathcal{N}(0,1).$$
>
> The following table shows the performance for different values of $\gamma$, showing improvements of up to 5 dB in reconstruction quality for 10 images of the test set of the Places365 dataset.
>
> | $\gamma$|0.0(Baseline)| 0.001| 0.01| 0.1| 1.0|
> |---|---|---|-----|----|------|
> | PSNR|17.59 ± 5.17| 18.79 ± 2.85| 21.93 ± 1.46| **22.16 ± 1.10**| 20.54 ± 2.06|
> | SSIM|0.311 ± 0.070| 0.277 ± 0.114| 0.488 ± 0.132| **0.542 ± 0.101**| 0.483 ± 0.171|
>
> [1] Solving Linear Inverse Problems Provably via Posterior Sampling with Latent Diffusion Models. NeurIPS 2023.
>
> [2] Solving Inverse Problems with Latent Diffusion Models via Hard Data Consistency. ICLR 2024.
>
> [3]  Diffusion Posterior Sampling for General Noisy Inverse Problems. ICLR 2023.
>
> [4] Denoising diffusion models for plug-and-play image restoration. CVPR 2023.
>
> [5] Deep image prior. CVPR 2018.

---

> > ### Comment · Reviewer_RKuN · 2025-08-02
> >
> > I thank the authors for answering my questions. The additional experiments strengthen the paper and I would encourage putting them in the future manuscript.
> >
> > I am still not sure how the model G is capable of estimating meaningful null-space projection. My confusion is that as G is a prediction network, it would output the conditional mean E_x P(Sx | Hx), which I would think would be close to zero, e.g.,  in the case of super-resolution (as x can vary widely).  In this case, why would the prediction help?

---

> > > ### Author Response · Authors · 2025-08-04
> > > **Response to reviewer**
> > >
> > > We thank the reviewer for the insightful comment. The method works when it is possible to find a non-linear correlation between null space (NS) components $\mathbf{Sx}$ and measurements $\mathbf{Hx}$ leveraging a dataset with triplets $(\mathbf{x}_i,\mathbf{Hx}_i, \mathbf{Sx}_i)$. We promote this correlation by solving Eq. (3) that balances the two terms: i) $\Vert\mathbf{Sx} - \mathrm{G}(\mathbf{Hx})\Vert$  encourages a non-linear correlation between $\mathbf{Sx}$ and $\mathbf{Hx}$. ii) $\Vert\mathbf{I} - \mathbf{A}^\top \mathbf{A}\Vert$ where $\mathbf{A} = [\mathbf{H}^\top, \mathbf{S}^\top]^\top$ promotes orthogonality and ensures that $\mathbf{S}$ samples components from the NS of $\mathbf{H}$.
> > >
> > > There are scenarios where the non-linear correlations are easier to achieve, for instance, in deblurring or SR, due to the low-pass filters associated with these tasks being non-ideal which leads to overlapping frequency bands between the high-pass components selected by $\mathbf{S}$ and the Gaussian-like low-pass components sampled by $\mathbf{H}$. In other scenarios, such as MRI or CT, the non-linear correlations between $\mathbf{Sx}$ and $\mathbf{Hx}$ are more challenging to achieve due to the orthogonality of their respective rows. But, the method still works by designing $\mathbf{S}$ such that it leverages structural similarities of  $\mathbf{H}$, such as adjacent frequencies in MRI or neighboring angles in CT.
> > >
> > > To illustrate the effectiveness of finding non-linear correlations between $\mathbf{Sx}$ and $\mathbf{Hx}$,  we set a challenging scenario in an MRI task with $\mathbf{H}$ sampling only random low frequencies using a 1D Cartesian mask with an acceleration factor (AF) of 12. We then apply an   _FFTSHIFT_ operation to this mask to produce a high-frequency sampling pattern, denoted by $\mathbf{S}_1$, spatially distant from the frequency support of $\mathbf{H}$ but maintaining the same AF. We refer to this configuration as the *well-separated sampling*. In this case, as the reviewer correctly pointed out, the method fails to recover non-trivial solutions, $\Vert\mathrm{G}^*(\mathbf{y})\Vert = 0.0108$, and a high relative error of $1.0$.
> > >
> > > Then, we configure a scenario where the non-linear correlations are easier to find, using the same $\mathbf{H}$ described before, we set $\mathbf{S}_2 = \mathbf{I} - \mathbf{H}$ that selects frequencies adjacent to those sampled by $\mathbf{H}$. This *adjacent sampling* configuration corresponds precisely to the setup employed in the MRI experiments presented in the main manuscript. It exhibits increased spectral (non-linear) correlation, resulting in a small relative error of 0.325, and a norm $\Vert\mathrm{G}^*(\mathbf{y})\Vert$ close to the norm of the ground-truth signal $\Vert\mathbf{Sx}\Vert$.
> > >
> > > We appreciate the reviewer for highlighting this limitation. In the final manuscript, we will explicitly discuss this failure case and clarify the conditions under which our method can or cannot recover meaningful NS components.
> > >
> > > |Sampling Type|$\Vert\mathbf{Sx}^{\*} - \mathrm{G}^{\*}(\mathbf{y})\Vert/\Vert\mathbf{Sx}^{\*}\Vert$|$\Vert\mathrm{G}^{\*}(\mathbf{y})\Vert$|$\Vert\mathbf{Sx}^{\*}\Vert$|
> > > |-|-|-|-|
> > > |Well-Separated Sampling|1.00|0.0108|1.72|
> > > |Adjacent Sampling|0.325|107|112|
> > >
> > > In the SR case mentioned by the reviewer, we want to show that in this case, $\mathrm{G}(\mathbf{y}) \neq 0$. As explained previously, this is because the non-linear correlations of $\mathbf{Hx}$ and $\mathbf{Sx}$ can be found easily (due to the overlapping of the blurring filter) such that $\mathbf{Sx}$ does not converge to the trivial solution $\mathrm{G}(\mathbf{y})=\mathbf{0}$. Here, the prediction of $\mathrm{G}$ provides components of the NS, i.e., high-frequency details of the image (similar spatial structure to $\mathbf{Hx}$).
> > >
> > > We used the setting as in the deblurring case, and set an SR factor $SRF=\sqrt{\frac{n}{m}}=4$. The forward operator is modeled as $\mathbf{H} = \mathbf{D}\mathbf{B}\in\mathbb{R}^{m\times n}$, where $\mathbf{B}\in\mathbb{R}^{n\times n}$ is a convolutional matrix with a Gaussian blur with bandwidth $\sigma=2.0$ and $\mathbf{D}\in\mathbb{R}^{m\times n}$ is the downsampling matrix.
> > >
> > > To recover the information removed by $\mathbf{H}$, we define the NS projection operator, similar to the deblurring case,  $\mathbf{S}[i,i+j] = 1 - \mathbf{h}[j], \quad i=1,\dots,m,\quad j=1,\dots,n,$, which captures the high-frequency details of the signal. We use the PnP-FISTA algorithm. In the following table, we show the reconstruction performance with baseline PnP-FISTA and NPN PnP-FISTA, showing an improvement of 1.74 dB in PSNR and 0.13 in SSIM. Additionally, we show the network estimation of the NS $\mathbf{Sx}$ metrics, showing good estimation of the high-frequency details from the low-resolution image.
> > >
> > > |Metric|PnP-FISTA Base|PnP-FISTA NPN|Estimation of $\mathbf{Sx}$|
> > > |-|-|-|-|
> > > |PSNR|22.01|23.75|24.95|
> > > |SSIM|0.562|0.692|0.697|

---

> > > > ### Comment · Reviewer_RKuN · 2025-08-05
> > > >
> > > > Thank you for the detailed explanation.  I understand the intuition now.  Perhaps what confused me is how the null space was referred to.  In the case of spatial-invariant blurring, the null space is the basis with eigenvalues equal to 0 -- ie, there is no overlapping between the null space and the measurement range space.  The authors should clarify this in the revision.

---

> > > > > ### Author Response · Authors · 2025-08-08
> > > > > **Concluding remarks to reviewer**
> > > > >
> > > > > Thank you for your constructive feedback on the paper. We will clarify these aspects in the final version of the manuscript.  Your comments have directly led us to strengthen the paper by adding new results on diffusion models, depeer analysis, and clarifications on the ability of network $\mathrm{G}$ to estimate $\mathbf{Sx}$ and the nature of the design of $\mathbf{S}$. We sincerely appreciate the time you invested and the thoughtful debate.

---

### Note · Authors · 2025-08-12

We thank the reviewers for the constructive dialogue. RKuN noted the clarity of our motivations, assumptions, and convergence analysis; hHJq emphasized NPN’s novelty beyond range–null decompositions and its relevance to ML and signal processing; PAt7 valued the focus on null-space (NS) structure. Here, we summarize the main improvements of the paper.

- Theorem 1 & CIZ: We clarified the guarantee of linear residual decay within a finite Convergence Improvement Zone (CIZ); outside it, standard PnP linear convergence holds. The CIZ quantifies accelerated decay, is non-empty, and practically reachable. We added matched-iteration summaries and corrected figure annotations, showing improved convergence ratios in deblurring, CT, CS, and MRI. Additionally, we noted that the RIP used holds widely (e.g., CS, MRI, certain deblurring/inpainting) and clarified the construction of matrix $\mathbf{S}$ (full-row rank, application-dependent), the role of $p$ and initialization, and when data distribution helps.

- Training of $\mathrm{G}$: Following the comments of reviewer RKuN, we detailed that $\mathrm{G}$ is trained in a supervised manner with orthogonality/invertibility regularizers to prevent trivial solutions. We provided intuition for when non‑linear correlations exist (e.g., deblurring/SR) and when they may be weak (well‑separated MRI sampling), which we state explicitly as a limitation, as performance is decreased in this case. We added experiments on SR showing consistent reconstruction performance and good estimation of NS components.

- Baselines & Comparisons: Following comments of reviewers PAt7 and RKuN, we added PnP-ADMM and diverse denoisers, observing consistent gains with NPN. Integrated into diffusion models (DM) for CT, NPN yields up to +1.85 dB (DPS) and +0.61 dB (DiffPIR); with Deep Image Prior (DIP), up to +4.5 dB. Per hHJq, we evaluated parameter-matched models: +1.57 dB over PnP-ADMM with only ~0.2 s extra; vs. an alternative learned regularizer of equal size, NPN is ~0.28 s faster and +2.67 dB better.

The reviewers concurred that our responses resolved their concerns. We appreciate their assessment and interest, which led to substantive improvements in the manuscript. The method is thoroughly validated across a wide range of applications, including CT, MRI, SR, CS, deblurring, and diverse solvers such as PnP (FISTA, ADMM), diffusion models, unrolling, and DIP with extensive simulations that substantiate the theoretical analysis.

---

### Decision · Program_Chairs · 2025-09-17

**Decision:**

Accept (poster)

**Comment:**

All three reviewers who provided a full review recommend acceptance (two borderline). The paper proposes a novel regularization technique for imaging inverse problems by focusing on the null space of the sensing operator. The reviewers praised the clear motivation, theoretical analysis, and the novelty of the approach. The main weaknesses identified were related to the clarity of some theoretical aspects and the need for more extensive comparisons, but the authors' rebuttals successfully addressed these concerns.